# How meteorological conditions influence aerosol-cloud interactions under different pollution regimes

Jianqi Zhao[1][2], Xiaoyan Ma[1], Johannes Quaas[2] and Tong Yang[1]

[1]China Meteorological Administration Aerosol-Cloud and Precipitation Key Laboratory, Nanjing University of Information Science and Technology, Nanjing 210044, China

[2]Leipzig Institute for Meteorology, Leipzig University, Leipzig, Germany

Correspondence: Xiaoyan Ma (xma@nuist.edu.cn)

**Abstract.** Aerosol-cloud interactions (ACI) involve complex physical and dynamical mechanisms, in which meteorological conditions play a crucial role. To investigate how the meteorological conditions impact ACI under different pollution regimes (polluted and clean) for marine liquid-phase clouds, the simulations are conducted using the chemistry version of Weather Research and Forecasting Model coupled with spectral-bin cloud microphysics. Our results indicate that, aerosols prolong cloud lifetime in humid environments. In dry conditions, they shorten cloud lifetime under low-to-moderate lower tropospheric stability (LTS) but further suppress the negative effects of entrainment in high-LTS environments with weak entrainment, ultimately extending cloud lifetime. Aerosols generally suppress precipitation but can enhance it during some intense cloud processes by promoting cloud vertical development in low-level clouds and increasing droplet amount in high-level clouds. The coordinated variations of cloud properties with meteorological conditions and aerosols reveal the regulatory role of meteorological factors on ACI. Specifically, under the winter monsoon background, the qualitative influence of aerosols on cloud properties is modulated by variations in updrafts and cold advection (characterized by LTS) driven by atmospheric circulation that co-varies with aerosol concentrations. Meanwhile, changes in atmospheric moisture govern the quantitative effects of aerosols on cloud development.

## 1 Introduction

Aerosol-cloud interactions (ACI) remain one of the major sources of uncertainty in climate assessment and prediction (IPCC, 2023). The complexity of aerosol impacts on clouds under varying meteorological conditions (Ma et al., 2018; Jia et al., 2019a; Anwar et al., 2022), along with difficulties in assessing these effects (Lohmann and Feichter, 2005; McComiskey and Feingold, 2012), constitutes a dominant contributor to uncertainty in ACI.

Liquid-phase clouds are the main contributors to the total impact of aerosols on clouds due to their broad coverage, high optical thickness, and often long lifetime (Chen et al., 2011; Andersen et al., 2017; Jia et al., 2019b). In addition, this type of cloud is highly sensitive to aerosol (Chen et al., 2014; Jia et al., 2019b). In liquid-phase clouds, ACI are primarily based on two mechanisms: (1) the Twomey effect, i.e., the entry of additional aerosol into the cloud leads to an increase in the cloud droplet number concentration ($N_d$), a decrease in the cloud droplet effective radius (CER), and an increase in cloud albedo under constant cloud water content (Twomey, 1977). (2) Rapid adjustments, consist of the response of cloud liquid water path (CLWP) and cloud fraction to changes in $N_d$ via the Twomey effect (Albrecht, 1989; Haghighatnasab et al., 2022; Jia et al., 2022). The effects of these mechanisms are modulated by meteorological conditions. Observation-based studies using ground-based measurements indicate that as boundary layer moisture increases, the impact of aerosols on cloud microphysics becomes more pronounced (Zheng et al., 2022). Analyses of satellite and reanalysis data reveal a positive correlation between CER and both lower tropospheric stability (LTS) and relative humidity (RH) over the East China Sea (Liu et al., 2024). Updraft is one of the primary drivers of ACI (Salma et al., 2021), but its influence varies in different environments. For instance, Liu et al. (2024) identified a negative correlation between CER and pressure vertical velocity over the East China Sea. Hudson and Noble (2014) demonstrated the sensitivity of ACI to updraft under varying aerosol number concentrations ($N_a$). Furthermore, large-scale circulation also plays a significant regulatory role in ACI (Dagan et al., 2023). Under the combined influence of these

meteorological factors, ACI form a complex nonlinear system. Observational analysis of this system suffers severely from retrieval uncertainties (Gryspeerdt et al., 2016; Arola et al., 2022), instrument errors (Eck et al., 2010; Allen et al., 2019), and interference from the radiative effects of aerosols and clouds (Sakaeda et al., 2011; Zelinka et al., 2014), thus requiring integration with numerical simulations.

This study employs the WRF-Chem-SBM model (Gao et al., 2016), which couples the online-chemistry version of the Weather Research and Forecasting model (WRF-Chem) with the spectral bin cloud microphysics (SBM) scheme, to analyze the sensitivity of ACI to meteorological conditions in different environments. In terms of aerosol-chemistry, compared with physics-only perturbation of aerosol, this model enhances the modeling capability of aerosols through the online treatment of aerosol properties (such as concentration, size distribution, and chemical composition), aerosol-related physical processes (such as emission, deposition, and thermodynamic and dynamic changes induced by aerosols), and chemical processes (such as gas-particle transition, heterogeneous reactions, and photochemical reactions) (Zaveri et al., 2008; Sha et al., 2022). This effectively reduces the distortion of ACI signals caused by inaccurate aerosol treatment (Ahmadov, 2016; Briant et al., 2017; Hodzic et al., 2023) and compensates for the lack of verifiability in physics-only perturbation of aerosol. In terms of cloud microphysics, compared with bulk, which cannot guarantee the accuracy of ACI signals due to (1) difficulties in accurately handling the CCN budget (Fan et al., 2012), (2) reduced sensitivity to aerosols resulting from the adoption of the saturation adjustment approach, and (3) the simplified treatment of the conversion from cloud water to rainwater and of hydrometeor fall velocities (Khain et al., 2015; Fan et al., 2016), the SBM used in this model, although computationally expensive and usually applicable only to relatively small domains for short time periods, and its accuracy being constrained by the theoretical understanding of cloud microphysics, is physically more realistic. Relevant evaluations have also demonstrated that it outperforms bulk in reproducing cloud–rain structures and resolving ACI (Khain et al., 2015).

The Eastern China Ocean (ECO), adjacent to Eastern China (EC)—one of the world's major anthropogenic emission centers—receives abundant aerosols from the continent. Moreover, marine clouds in this region are less affected by complex land surface processes (Zhao et al., 2024), allowing ACI to be examined more clearly than over terrestrial areas. Therefore, this study selects ECO as the research domain. Previous ACI modeling studies, conducted both over ECO (Saleeby et al., 2010; Bennartz et al., 2011) and in other regions (Liu et al., 2020; Guo et al., 2022; Haghighatnasab et al., 2022), have provided extensive quantitative and qualitative insights into ACI, greatly advancing our understanding of aerosol impacts on clouds and the factors influencing ACI. Building on this foundation, our previous work (Zhao et al., 2024) evaluated the performance of WRF-Chem-SBM in simulating aerosol–cloud interactions over both EC and the ECO during a winter case characterized by a high frequency of low clouds (Chang et al., 2021; Niu et al., 2022). That study refined the understanding of ACI mechanisms and influencing factors over EC and ECO, and demonstrated the land–sea contrasts. Following this work, the objective of the present study is to further elucidate the complex and nonlinear meteorology–aerosol–cloud system by systematically analyzing how meteorological conditions influence ACI under different environments. Moreover, to address several common limitations in ACI modeling efforts—such as (1) interference of external factors (e.g., aerosol and cloud radiative effects) with ACI signals, (2) the limited generality of case-specific quantitative and qualitative conclusions, and (3) uncertainties in signal separation due to co-varying influences among factors—this study introduces improvements in modeling configuration (four-dimensional assimilation and disabling radiative effects), experimental design (selection of a case featuring rich meteorological-aerosol-cloud variations and a large inter-experiment aerosol difference), and analysis approach (examination of ACI responses to the co-variation of different meteorological factors and of meteorological factors and aerosols), as detailed in Section 3.3.

This paper is structured as follows: Section 2 introduces the model used in this study, the observational data, and the data processing methods. Section 3 describes the case analyzed in this study and evaluates the simulation results. Section 4 presents the analysis and discussion of ACI based on simulation results. Section 5 provides the summary of the study.

# 2 Methods and data

## 2.1 Model introduction

This study utilizes the WRF-Chem-SBM model (Gao et al., 2016), which couples the Model for Simulating Aerosol

Interactions and Chemistry (MOSAIC) aerosol module (four-bin) of the WRF-Chem model and the SBM scheme, to perform aerosol-cloud simulation. The MOSAIC aerosol module treats the mass and number distributions of the nine main aerosol species (sulfate, nitrate, chloride, ammonium, sodium, black carbon, primary organics, other inorganics, and water). The aerosol particles are assumed to be internally mixed (Zaveri et al., 2008). The diameters of the four bins range from 0.039-0.156, 0.156-0.624, 0.624-2.5 and 2.5-10.0 μm, respectively. The module treats processes such as aerosol emissions, gas-particle transition, coagulation, in-cloud liquid-phase chemistry, dry deposition, and wet scavenging (Sha et al., 2019; Sha et al., 2022). The SBM scheme in this model is the fast version, which solves a system of prognostic equations for three hydrometeor types (droplets, ice/snow, and graupel) and CCN, by numerically discretizing their size distributions. Each size distribution function is structured into 33 mass-doubling bins, wherein the mass within the k-th size bin is twice that within the (k-1)-th bin. This scheme can treat cloud microphysics processes such as aerosol activation, freezing, melting, diffusion growth/evaporation of droplets, deposition/sublimation of ice particles, and droplet and ice collisions (Khain et al., 2004).

In this coupled model, aerosols from MOSAIC are distributed into 33 CCN bins based on their physical and chemical properties and are then processed through the aerosol activation, resuspension, and in-cloud wet removal parameterizations in SBM, which not only impact cloud microphysics but also provide feedback to the aerosol module. The aerosol activation parameterization in this model is based on the Köhler theory, which calculates the critical supersaturations ($S_{crit}$) at the boundary sizes for each CCN bin. When the prognostic supersaturation falls between the boundary-size $S_{crit}$ of a given bin, the CCN in that bin are fractionally activated. CCN in larger bins with lower $S_{crit}$ are completely activated, while CCN in smaller bins do not undergo activation (Gao et al., 2016).

2.2 Experiment setup

In the simulations, nested grids with 48 vertical layers and horizontal resolutions of 12 km and 2.4 km are employed. The outer domain (all colored areas in Fig. 1, including the parts obscured by the Reference Vector) is centered at (32°N, 120°E) with a grid number of 151×125, while the inner domain covers ECO (highlighted by the red box in Fig. 1, it spans longitudes 123.03–126.24°E and latitudes 30.41–33.14°N) with a grid number of 121×121. The simulation period is from 00:00 on Feb. 1, 2019, to 00:00 on Feb. 5, 2019, in UTC. The initial 12 hours are considered as model spin-up and are not included in the analysis. The model outputs once per hour.

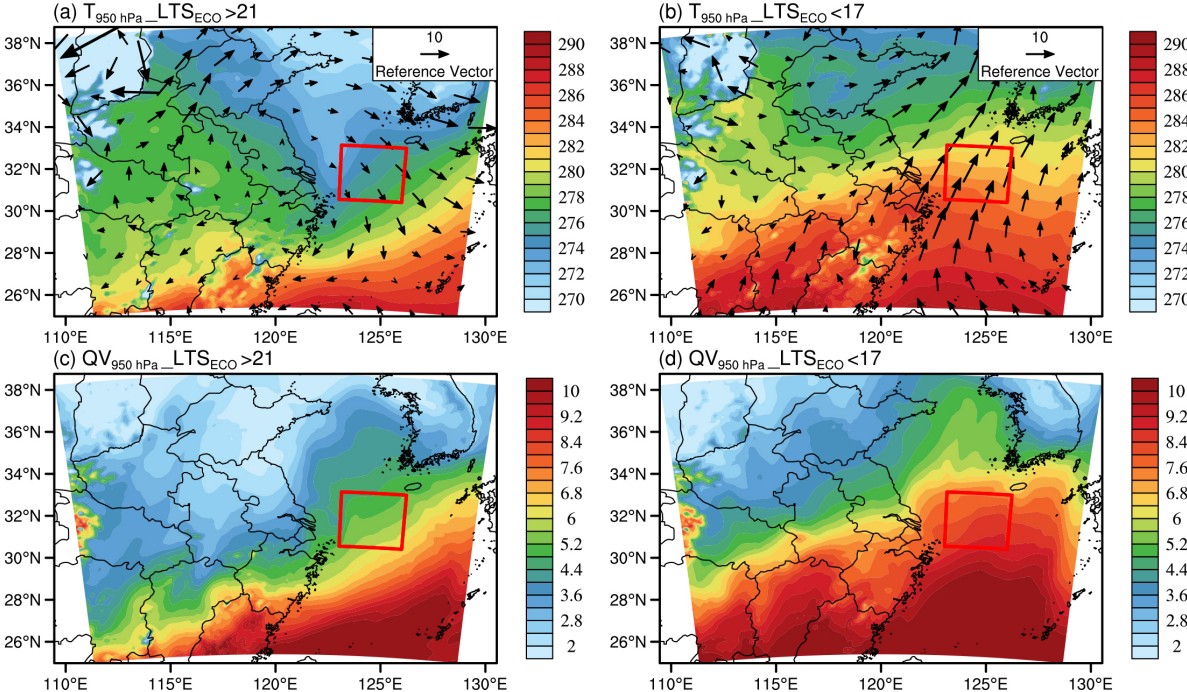

**Figure 1.** The 950 hPa temperature (a-b, in K), wind field (a-b, in m·s⁻¹) and water vapor content (c-d, in g·m⁻³) averaged over periods when the LTS in ECO (highlighted by the red box) is high (>21, a and c) and low (<17, b and d), from the Control

experiment.

Meteorological initial and boundary conditions are from the National Center for Environmental Prediction (NCEP) FNL reanalysis with a temporal and spatial resolution of 6 h and 0.25°, respectively (NCEP et al., 2015). Chemical initial and boundary conditions are from the Community Atmosphere Model with Chemistry (CAM-chem, Buchholz et al., 2019; Emmons et al., 2020). Anthropogenic emissions are from the Multi-resolution Emission Inventory for China (MEIC, 2016 version developed by Tsinghua University (Li et al., 2017; Zheng et al., 2018). To avoid interference from aerosol and cloud radiative feedback on ACI signals, aerosol and cloud radiative effects are disabled in the simulations (a more detailed discussion is provided in Section 3.3). The model parameterization settings for the Control experiment are listed in Table 1. In the Clean experiment, continental emissions (including anthropogenic, dust, and biogenic sources) as well as aerosol and chemical initial and boundary conditions from CAM-chem are disabled.

**Table 1.** Model parameterization settings for the Control experiment. "Number" refers to the WRF-Chem namelist switch.

| Process | Number | Name |
|---|---|---|
| Cloud microphysics | 30 | SBM (Khain et al., 2004) |
| Longwave radiation | 4 | RRTMG (Mlawer et al., 1997) |
| Shortwave radiation | 4 | RRTMG (Iacono et al., 2008) |
| Surface layer | 1 | MM5 Monin-Obukhov (Pahlow et al., 2001) |
| Land surface | 2 | Unified Noah (Chen et al., 2010) |
| Boundary layer | 1 | YSU (Shin et al., 2012) |
| Cumulus | 5 | Grell-3 (Grell et al., 2014) |
| Chemistry and aerosols | 9 | CBMZ and four-bin MOSAIC (Sha et al., 2022) |
| Photolysis | 2 | Fast-J (Wild et al., 2000) |
| Sea salt emission | 2 | MOSAIC/SORGAM (Fuentes et al., 2011) |
| Dust emission | 13 | GOCART (Zhao et al., 2010) |
| Biogenic emission | 3 | MEGAN (Guenther et al., 2006). |

In addition, the four-dimensional assimilation method is used in this study to improve the model's ability to simulate the meteorological field and thus enhance the model's ability to reproduce the factual aerosol-cloud scenario (Zhao et al., 2020; Hu et al., 2022). This method incorporates relaxation terms derived from model errors at observational stations, which act to systematically reduce those errors (Liu et al., 2005). Each observation is assigned a user-defined radius of influence, a time window, and a relaxation time scale—collectively determining where, when, and to what extent the observation adjusts the model solution. Model grid cells often fall within the influence ranges of multiple observations, and in such cases, the contributions are weighted based on their distance from each observation. The assimilation uses observations of altitude, wind direction, wind speed, air pressure, temperature, and dew point, obtained from the NCEP operational global surface (NCEP et al., 2004) and upper air (Satellite Services Division et al., 2004) observation subsets.

2.3 Observational data

In order to evaluate the simulated meteorological conditions, aerosols, clouds, and precipitation, various observational datasets are utilized, including height, temperature, dew point depression, and wind field (at 11 pressure layers, 12 hours temporal and 2.5° spatial resolutions) from the Meteorological Information Comprehensive Analysis and Processing System (MICAPS, Hu et al., 2018) developed by the National Meteorological Center of China, hourly near-surface $PM_{2.5}$ data from more than 1600 stations provided by the National Urban Air Quality Real-time Release Platform (China National Environmental

Monitoring Center, 2023), AOD from the MOD04_L2 dataset (Levy et al., 2015) which combines the "Dark Target" and "Deep Blue" algorithms with temporal and spatial resolutions of 5 min and 10 km, as well as cloud properties (such as CER, cloud optical thickness (COT), cloud top height (CTH), cloud top temperature (CTT), and CLWP) from the MOD06_L2 dataset (Platnick et al., 2015) with temporal and spatial resolutions of 5 min and 1 km. The column-integral $N_d$ is calculated from COT and CER based on the approach of Han et al. (1998, see also Brenguier et al. (2000)):

$$N_d = \gamma \cdot COT^{0.5} \cdot CER^{-2.5} \tag{1}$$

where $\gamma$ is a constant valued at $1.37 \times 10^{-5}$ m$^{-0.5}$ (Quaas et al., 2006). Compared to in-situ observations, the $N_d$ retrievals perform well for relatively homogeneous, optically thick and unobscured stratiform clouds under high solar zenith angle conditions (Grosvenor and Wood, 2014; Bennartz and Rausch, 2017; Jia et al., 2021). However, at low COT, the retrievals suffer from large uncertainties arising from instrument errors and other sources of reflectance error for COT and CER (Zhang and Platnick, 2011; Sourdeval et al., 2016). This is the primary rationale for excluding MODIS transparent-cloudy pixels (COT < 5), as described in Section 2.4. In addition, the 5-min temporal resolution of MOD04_L2 and MOD06_L2 data originates from the satellite's scanning frequency. However, since the Terra satellite passes over a fixed location only once per day (it passes over ECO at approximately 03:00 UTC, which corresponds to around 11:00 local time), the number of valid data for a given location within ECO typically does not exceed five per day (Terra satellite scans across adjacent time periods may cover the same region).

Precipitation data is obtained from the Integrated Multi-satellitE Retrievals for GPM (IMERG) dataset (Huffman et al., 2019), of which the daily accumulated high quality precipitation product is used in this study, with a temporal and spatial resolution of 1 day and 0.1°, respectively.

2.4 Data processing

Due to the resolution differences between the observations and simulations, interpolation is needed before comparing them. It follows the principle of interpolation from high resolution to low resolution. Specifically, for observational data with higher resolution than the model grid, the observational data is interpolated to the model grid, while for observational data with lower resolution than the model grid, the simulation data is interpolated to the grid of the observational data.

In addition to the above, further processing is required for both satellite-retrieved and simulated liquid-phase cloud properties. For MODIS cloud properties, cloud retrievals with high reliability are selected using the method referenced from Saponaro et al. (2017): (1) Liquid-phase cloud data are chosen based on MODIS cloud phase parameters, and (2) transparent-cloudy pixels (COT < 5) are screened out to limit uncertainty (performing this screening results in the inability to evaluate simulations with low COT pixels, reducing the completeness of the evaluation but increasing its reliability). When calculating the simulated cloud properties for a specific grid cell, in cases where multiple independent clouds exist at different altitudes, we follow the method of the instrument simulator, treating these clouds as if they are from a single homogeneous layer (Pincus et al., 2012). When comparing the simulations with satellite-retrieved cloud properties, we refer to the processing of satellite-retrieved cloud properties and the threshold method used by Roh et al. (2020) for distinguishing cloud phases. The following criteria are applied to filter the simulated cloud properties: (1) cloud optical thickness of water (COTW) > 0.1 and cloud optical thickness of ice (COTI) < 0.01 for each layer, and (2) column COTW ≥ 5. The highest layer meeting this condition is identified as the cloud top. The simulated COTW and COTI used in this study are calculated based on the methodology in the Goddard radiation scheme (Zhong et al., 2016) of the WRF-Chem model.

In the evaluation, the method used to calculate simulated $N_d$ is the same as that used for satellite-retrieved $N_d$. In the analysis of the simulation results, $N_d$ is directly taken from the model output, and the criteria for liquid-phase clouds are strictly defined as cloud liquid water content (CLWC) > 0 and cloud ice water content (CIWC) = 0 (the processing of the simulation results at evaluation is no longer used in the analysis). Based on the results of the Control experiment, we present the $N_d$ values derived from both the empirical formula calculation and the direct model output (Fig. S1). The model-output and empirically derived $N_d$ values exhibit broadly consistent trends with changes in COT and CER, as further demonstrated by the stable ratio of the empirically derived $N_d$ (after division by 10.0) to the model-output $N_d$. However, a systematic discrepancy exists: the in-cloud mean $N_d$ from direct model output is approximately one order of magnitude lower. This difference arises because the

model employs a more inclusive criterion for defining liquid cloud layers, incorporating numerous thin cloud layers (with CIWC = 0 and CLWC > 0) that are typically excluded by satellite retrievals. The inclusion of these thin layers in the vertical averaging process leads to a systematically lower average $N_d$. Furthermore, for optically thin clouds (COT < 5.0), the empirical formula shows a tendency to overestimate $N_d$ (Fig. S1c), a finding that aligns with the evaluation of satellite $N_d$ retrievals cited in Section 2.3 and is consistent with the rationale for the screening criteria of MODIS data. When analyzing simulated ACI signals, using $N_d$ calculated by Eq. (1) would lead to severe overestimation of $N_d$ for clouds with low $N_d$, introducing significant uncertainty in ACI analysis under low LTS and/or low $N_a$ conditions (Fig. S1f, i, and l). Yet, removing these optically thin clouds would compromise the comprehensiveness of the analysis. Therefore, in our ACI analysis based on simulation results, we use model-output $N_d$ that is directly correlated with simulated aerosols, CLWP, and precipitation—a level of consistency that cannot be robustly achieved using the empirical formula.

When comparing the spatial distribution of simulation results with satellite-retrieved data averaged over the study period, it is unreasonable to directly compute the average of the simulation results for the entire period due to the spatio-temporal discontinuities of satellite-retrieved data. Therefore, to make the comparison with the satellite-retrieved data, spatio-temporal matching is performed for the simulation data. Specifically, only when satellite data at a given grid cell and time is valid, the corresponding model output (at the nearest time step to the satellite data at the same grid cell) is included in the average calculation. During the study period, the number of valid MODIS records varied across grid cells, with the averaged value for each cell corresponding to 0 to 6 records.

It should be noted that the mean value of CER is easily disturbed, particularly by extremely high values under low-$N_d$ conditions, which may hinder analyses of aerosol-cloud relationships based on mean statistics. To address this, the mean CER shown in the figures of this study (which only include simulated CER, and the evaluation of simulated CER is incorporated into $N_d$ derived from CER and COT) is calculated by first summing the total number of droplets in each size bin over all samples (across space, time, and vertical levels), and then applying the following formula:

$$CER = \frac{\sum_i r_i^3 N_i}{\sum_i r_i^2 N_i} \tag{2}$$

where $r_i$ is the radius of the $i$-th bin of cloud droplets and $N_i$ is the corresponding total number of droplets.

We employ multiple statistical metrics to support the analysis. Data uncertainty is quantified using the 25th-75th percentile range, mean, median, 1.5 interquartile range (IQR), and outliers. Pearson correlation is used to assess relationships between variables. For datasets with complete spatiotemporal coverage, correlations are computed at each grid cell. In the case of multi-layer datasets, values are interpolated to common vertical levels and reshaped into one-dimensional arrays ordered from bottom to top. For datasets with limited spatiotemporal completeness, we apply the uncentered Pearson product-moment coefficient to evaluate spatial correlations of grid cell means. Relative differences between parameters are assessed using the Normalized Mean Bias (NMB).

# 3 Experiment design and model evaluation

### 3.1 Case description

This study aims to investigate how the effects of continental aerosols on liquid-phase clouds and ACI under different pollution regimes respond to meteorological conditions over ECO. For this purpose, we (1) design two experiments—Control (polluted regime, including both continental and marine emissions) and Clean (clean regime, including only marine emissions)—to qualitatively and quantitatively analyze the sensitivity of ACI to meteorological conditions in different environments, and (2) select a short-term case characterized by substantial variability in meteorological, aerosol, and cloud properties (liquid-phase clouds over ECO from February 1 to 5, 2019), balancing computational cost with analysis comprehensiveness.

Fig. 2 presents the temporal evolution of LTS, column-mean RH from the surface to 1300 m (this range captures over 97%

of cloud droplets in both experiments and is used for all meteorological variables to better reflect their influence on ACI), aerosol number concentration ($N_a$), and CLWP over ECO from the two experiments. Changes in atmospheric circulation alter the meteorological conditions over ECO, as characterized by variations in LTS. When LTS is low (Figs. 1b and S2), ECO is primarily influenced by southerly winds, under which the lower atmosphere exhibits characteristics of warmth, moisture, and dominant updraft. As LTS increases (Fig. 1a and S2), ECO becomes dominated by dry, cold northwesterly winds, and both temperature and water vapor content in the lower atmosphere gradually decrease. Statistical analysis of grid cells over ECO reveals a clear negative correlation between temperature and water vapor content at 950 hPa and LTS (correlation coefficients are -0.68 and -0.58, respectively, with p-values both less than 0.01). Accompanied by large-scale subsidence from the northwestern high-pressure system, a significant capping inversion forms at the top of the ECO boundary layer, strongly inhibiting updraft (Fig. S2d and i). However, due to intense condensation caused by the advection of northwestern low-level cold air, the lower atmosphere over ECO exhibits higher supersaturation than under low LTS (Fig. S2e and j).

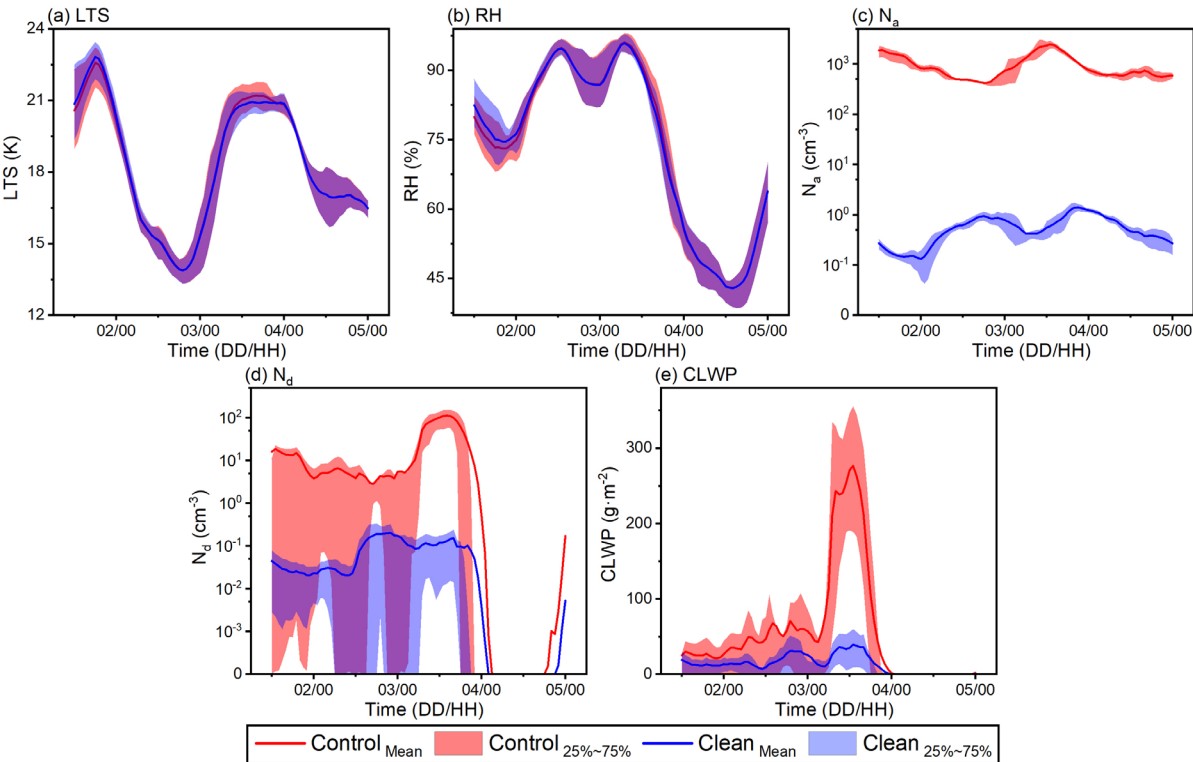

**Figure 2.** Variations in LTS (a), column-mean RH (b) from the surface to 1300 m, in-cloud mean $N_a$ (c) and $N_d$ (d), and CLWP (e) averaged over ECO during the study period for the Control and Clean experiments.

For aerosols, the average $N_a$ under polluted and clean regimes differs by a factor of 1988. Under the polluted regime, ECO experiences high-pollution periods ($N_a > 2400$ cm$^{-3}$, located along the major continental aerosol transport pathway) and relatively low-pollution periods ($N_a < 500$ cm$^{-3}$, away from the pathway). Under the clean regime, aerosol concentrations over ECO also exhibit significant variability driven by marine emissions, with $N_a$ ranging from 0.13 cm$^{-3}$ to 1.4 cm$^{-3}$, differing by more than an order of magnitude. $N_d$ and CLWP also exhibit pronounced variations in response to meteorological conditions and $N_a$.

As mentioned above, the shift in meteorological conditions over ECO during the simulation period, the distinct aerosol variations in each experiment, the significant aerosol differences between the two experiments, and the resulting disparities in cloud properties—both within each experiment and between them—collectively provide abundant samples for ACI analysis. This robustly supports the comprehensiveness of the study.

## 3.2 Model evaluation

Multiple observational datasets are used in this study to evaluate the simulated meteorological fields, aerosols, clouds, and precipitation. The simulated cloud and precipitation in the inner domain are compared with the observations, while the simulated meteorological fields and aerosols in the outer domain are used for comparison due to limitations in observational data resolution and availability. The vertical profiles of domain mean observed and simulated meteorological elements (Fig. 3) indicate that the simulations with (Control experiment) and without (Control_NoDA—the Control experiment without assimilation) assimilation both reasonably reproduce the vertical variation of the observations. From the perspective of vertical profiles, assimilation has enhanced the modeling of some meteorological elements, yet it also amplifies the simulation-observation discrepancies in some vertical layers. The improvement in meteorological simulations via data assimilation is clearly demonstrated by the higher simulation-observation correlation coefficient (Fig. S3). With the exception of a minor reduction in the correlation between simulated and observed height (a regional average decrease of merely $4.75 \times 10^{-7}$), most areas within ECO demonstrate significant improvements in simulation-observation correlations. The most notable enhancement occurs in simulation-observation correlations of dewpoint depression—critical for liquid-phase cloud simulation—with an average increase of 0.11 and a mean relative improvement of 16.02%. Beyond improving the meteorological simulation overall, assimilation also plays an important role in ensuring meteorological consistency between the two experiments and enhancing the robustness of simulated signals (see Section 3.3 for detailed explanations).

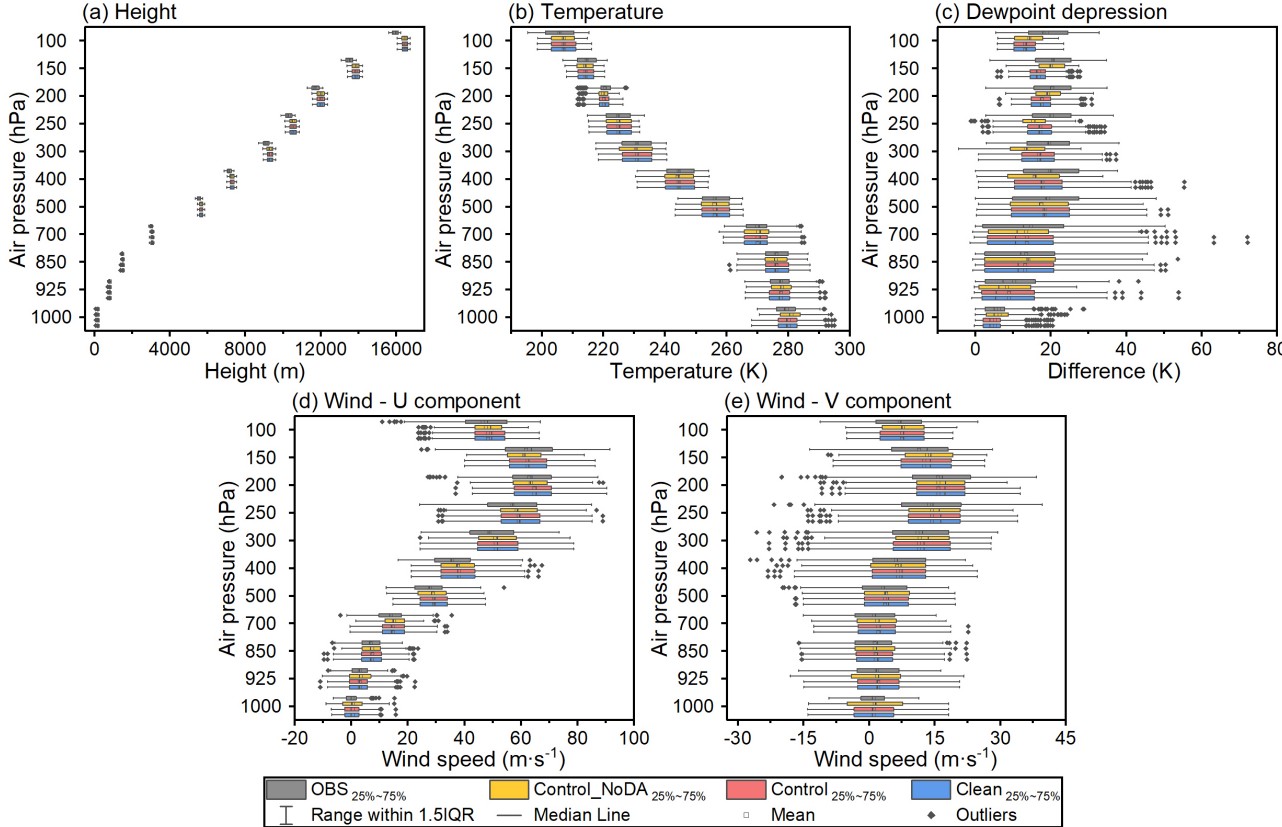

**Figure 3.** The observed (MICAPS, in black) and simulated (Control_NoDA—the Control experiment without assimilation, shown in orange, Control—the Control experiment with assimilation, shown in red, and Clean—the Clean experiment with assimilation, shown in blue) height (a), temperature (b), dewpoint depression (c), and zonal (d) and meridional (e) winds of the outer domain at each pressure level, including the 25th to 75th percentile range, mean, median, range with 1.5 IQR (Interquartile Range), and outliers.

Supported by realistically reproduced meteorological fields, high-resolution emission data, and well-defined aerosol initial and boundary conditions, the model accurately captures the spatial distribution of observed AOD and near-surface $PM_{2.5}$ (Figs. 4a-f), with spatial correlations of 0.80 and 0.95 and NMBs of -13.22% and 6.78%, respectively. Although direct $PM_{2.5}$ observations are unavailable over ECO, the consistency between AOD, $PM_{2.5}$ from nearby areas, and the accuracy of wind field simulations suggests that the model can reasonably reproduce aerosol characteristics in this region.

Compared to aerosol simulations, cloud and precipitation modeling remains more challenging due to uncertainties in microphysical processes (Figs. 4g-u). While clear discrepancies are found between simulated and observed $N_d$ and CLWP in northeastern and southern ECO, the model generally captures the distribution of MODIS $N_d$ and CLWP reasonably well, with correlations reaching 0.81 and 0.72, and NMBs of only -0.01% and 8.3%, respectively. For cloud-top parameters, although the model reproduces the spatial pattern of MODIS CTT—lower in the north and higher in the south—and achieves correlations of 0.83 and 0.99 with MODIS CTH and CTT, respectively, it systematically underestimates CTH by an average of 53.71% compared to MODIS. This discrepancy stems partly from simulation errors, as also indicated in $N_d$ and CLWP, but more significantly from differences in cloud-top detection methods and limitations in model parameterization. For cloud-top identification, MODIS employs the $CO_2$ slicing method (Platnick et al., 2015), while our simulation identifies cloud tops based on COTW and COTI. Furthermore, as shown in Fig. 4p-r and Fig. S4, MODIS-detected liquid-phase CTT can reach below 240 K (-33.15°C), whereas the simulated CTT in this study rarely falls below 266 K (-7.15°C). Under the model's parameterization, only mixed-phase or ice-phase clouds exhibit lower CTT values.

For precipitation, the model captures the observed precipitation zones in northwestern ECO and the southwest-northeast rainband, though slight spatial shifts result in a relatively low correlation (0.57) with observations. Overall, the model reproduces the spatial distributions of aerosol, cloud (apart from systematic errors in cloud top parameters), and precipitation parameters within an acceptable error range (±30%), and uncertainty statistics in Fig. S4 further support the consistency between simulations and observations. The evaluation provides a credible foundation for simulation-based analyses of ACI.

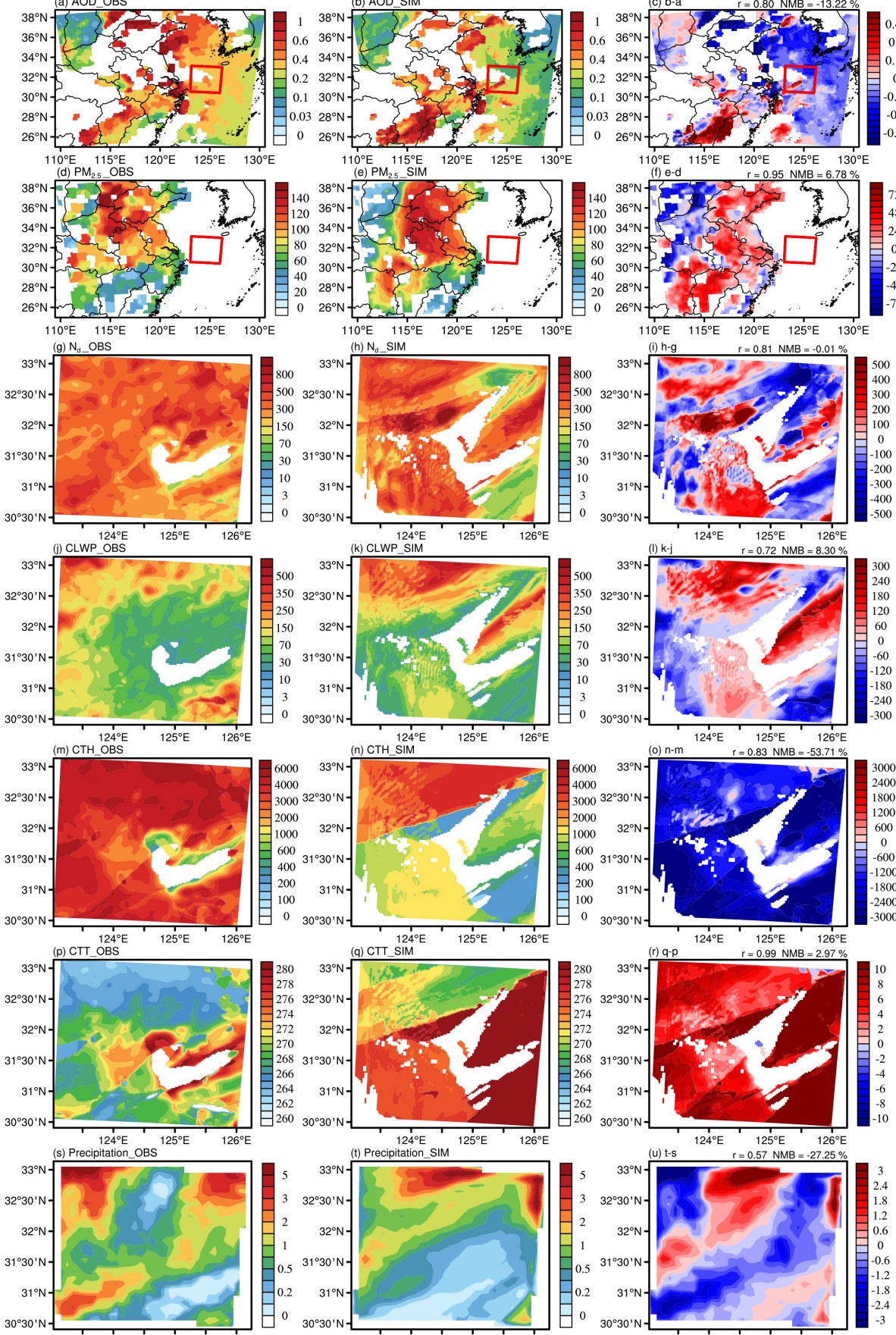

**Figure 4.** Averaged distributions of observed (left column. AOD and cloud properties, near-surface PM₂.₅, and precipitation from MODIS retrievals, near-surface observations, and IMERG, respectively) and simulated (middle column, from Control

305

experiment) AOD (a-c) and near-surface $PM_{2.5}$ (d-f, in $\mu g \cdot m^{-3}$) of the outer domain, and $N_d$ (g-i, in $cm^{-3}$), CLWP (j-l, in $g \cdot m^{-2}$), CTH (m-o, in m), CTT (p-r, in K), and cumulative precipitation (s-u, in mm, and as the IMERG resolution is 1 day, here is the cumulative precipitation for February 2-4) of the inner domain (the areas within the red boxes in Figs. a-f), as well as the differences between simulations and observations (right column, r and NMB in the upper right corner represent the spatial correlation coefficient and normalized mean bias between simulation and observation). The simulated data in the figure are processed using interpolation and spatio-temporal matching (the latter is applied only for comparison with satellite-retrieved data exhibiting spatio-temporal discontinuities), as detailed in Section 2.4. For each grid cell, the corresponding simulated result is displayed only when observational data is available.

3.3 The robustness of the simulated signals

The robustness of the signal serves as the foundation for analyzing ACI based on simulation results. Its key lies in the accuracy of ACI signal separation and the control of simulation errors. The main challenge hindering ACI signal separation stems from extensive external interference, which specifically includes: (1) aerosol-cloud feedback (e.g., radiative effects of aerosols and clouds altering the atmospheric thermal structure, and wet deposition feedback on ACI), (2) covariation between meteorological fields (such as humidity, vertical motion, and atmospheric stability) and aerosols (e.g., simultaneous occurrence of meteorological and aerosol conditions favorable for cloud development may confound the attribution of cloud enhancement), and (3) underlying surface influences (e.g., effects of the surface on radiation and water vapor, as well as topographic uplift). To mitigate these confounding factors and improve the accuracy of ACI signal separation, efforts have been made in experimental design, simulation techniques, and analytical methods.

Aerosol-cloud feedback can confound the ACI signal. First, the radiative effects of aerosols and clouds exert significant impacts on both ACI simulations and signal separation by altering atmospheric temperatures (Wall et al., 2022; Dagan et al., 2023). We have conducted tests (all tests use settings consistent with the Control experiment except for radiation settings) with cloud radiative effects enabled (Rad_CLD), aerosol radiative effects enabled (Rad_AER), and both enabled simultaneously (Rad_ALL), comparing these results with those from the Control experiment where both effects are disabled (Fig. S5). From the regional average perspective, the radiative effects of aerosols and clouds show minimal influence on the temporal trends of meteorology, aerosols, and clouds. However, during specific periods, the radiative effects significantly alter Na by influencing transport and deposition processes and modify cloud properties by affecting meteorological and aerosol conditions. Their impact on the regional averages of $N_a$, $N_d$, and CLWP exceeds 20% at certain times. In terms of ACI signals, the cloud radiative effect accelerates the increase of $N_d$ with Na and the increase of CLWP with RH. In contrast, the aerosol radiative effect opposes the radiative effect of clouds. The mutual offset between these effects results in the Rad_ALL test exhibiting the closest agreement with the Control experiment for these two trends. Compared to these, the relationship between clouds and LTS exhibits weaker regularity. The perturbation from ARI further increases the uncertainty, posing a significant challenge to elucidating the influence of LTS on ACI. Overall, from the perspective of ACI analysis, the quantitative deviations and enhanced qualitative uncertainty caused by ARI necessitate turning off ARI in ACI simulations. Related studies also often use this setting to remove the impact of ARI (Liu et al., 2020; Wang et al., 2020). Regarding wet deposition feedback on ACI, directly turning it off would negatively impact ACI simulation and analysis. Hence, this study instead monitors its influence on ACI by synchronously tracking aerosol-cloud-precipitation variations.

This study primarily controls external factor (2) through experimental design, technical methods, and analytical approaches. First, as previously mentioned in case selection, the strong variations in aerosols and meteorological fields of the simulations provide relatively abundant samples for separating ACI under the interference of aerosol-meteorology covariation. Second, technically, disabling aerosol and cloud radiative effects, along with four-dimensional data assimilation, supports consistency in meteorological fields between the two experiments. Vertical profiles (Fig. 3) and the spatial distribution of inter-experiment correlations (Fig. S6) of meteorological parameters reveal highly consistent patterns and strong correlations (with correlations at every grid cell exceeding 0.99, all passing the 99% significance test, and the average absolute NMB for all variables being only 0.06%) between the two experiments. Under identical meteorological conditions, comparisons between the two

experiments effectively avoid the influence of meteorological covariation. Third, the analysis of this study demonstrates the two-dimensional variations of meteorology-aerosol-cloud-precipitation with respect to changes in aerosols and meteorological fields, providing direct support for assessing the sensitivity of ACI to meteorology and aerosols. Additionally, regarding external factor (3), selecting the study region over ocean rather than continental areas effectively mitigates this influence.

In addition to the errors between the Control experiment and observations discussed in Section 3.2, another key aspect of controlling simulation errors lies in ensuring the robustness of the inter-experiment differences—specifically, by guaranteeing that the differences between the two experiments exceed the simulation error between the Control experiment and observations. Beyond the technical measures mentioned earlier, we demonstrate this point from two perspectives: experimental design and statistical analysis of aerosol and cloud properties. First, the only difference between the Control and Clean experiments is using continental aerosols. According to the simulation results, the average aerosol number concentration in the two experiments differs by a factor of 1988, representing an extreme forcing scenario and providing a strong foundation for the analysis. This strong external perturbation inherently suggests that if aerosols have an impact on clouds and precipitation, it should be reflected in the simulation. Second, the statistical analysis of the uncertainty in aerosol and cloud properties between the observations and simulations, shown in Fig. S4, indicates that the differences in aerosol and cloud properties between the two experiments are much larger than those between the Control experiment and the observations (except for differences in cloud top parameters caused by systematic errors). Due to the offsetting effects of aerosols on both enhancing and weakening precipitation (Figs. 4s-u), the 25%-75% range of the two experiments is similar (Fig. S4g). However, in the range with 1.5 IQR and outliers, the differences between the two experiments are comparable to the differences between the Control and observations, providing support for the significance of the results.

These efforts contribute to improving the accuracy of ACI simulation and the clarity of signal separation, thereby supporting the reliability of subsequent ACI analysis.

# 4 Results and discussion

4.1 General variation of ACI with meteorological conditions

Continental aerosols exert a substantial impact on liquid-phase clouds over ECO (Fig. 5) due to their vast quantity (reaching 1,988 times that of marine emissions). This leads to an average 248-fold increase in $N_d$, a 77.74% decrease in CER, and a 2.75-fold rise in CLWP across this region. Compared to their effects on clouds, the influence of continental aerosols on precipitation is more heterogeneous, with both enhancements and suppressions partially offsetting each other, resulting in an overall 6.86% increase in precipitation over ECO. Based on differences in cloud properties between the Clean and Control experiments—driven by continental aerosols—and their variations with meteorological conditions (using LTS to represent dynamic changes and RH to represent atmospheric moisture variations), we analyze the overall influence of meteorological conditions on ACI.

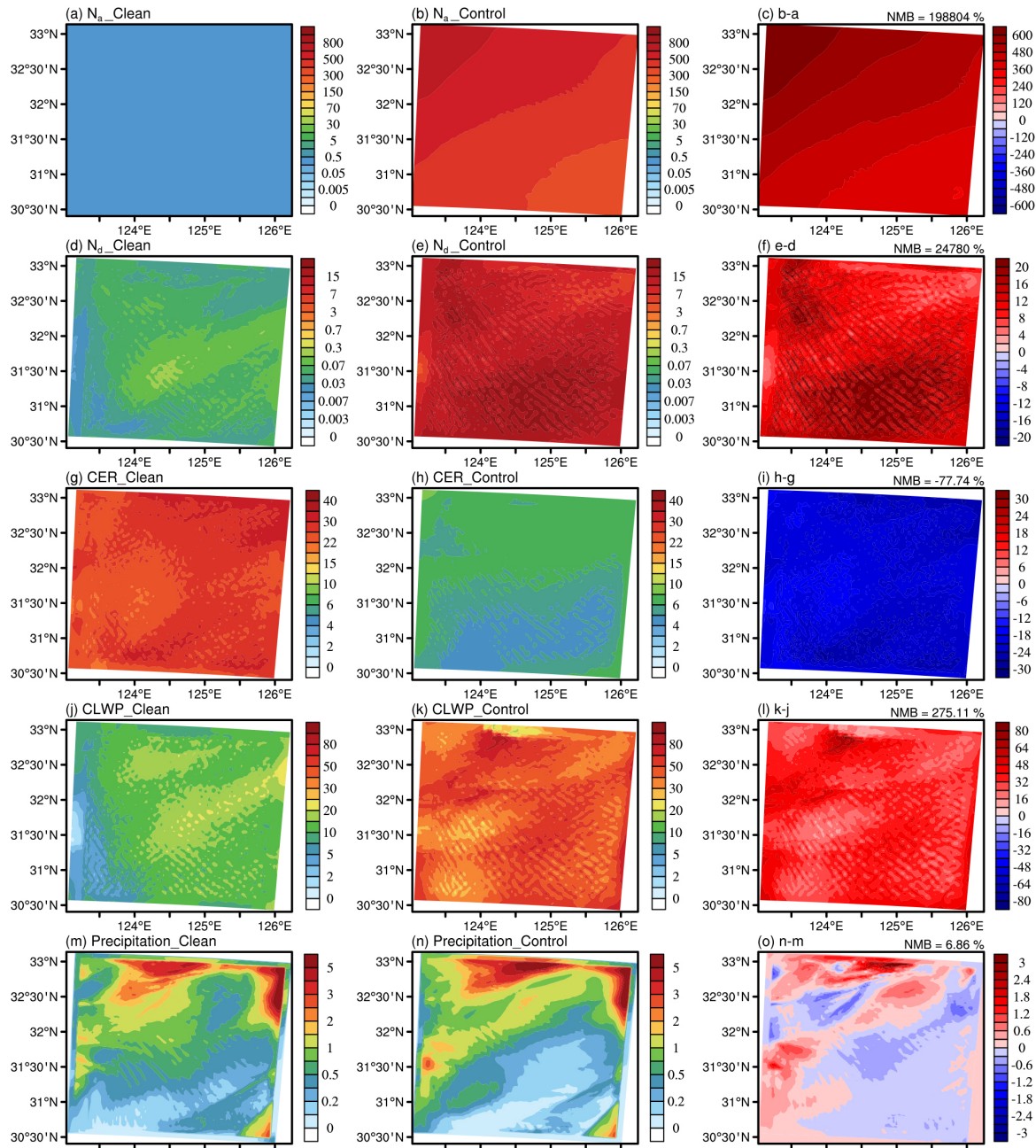

**Figure 5.** Column-mean $N_a$ (a-c, in cm$^{-3}$), in-cloud mean $N_d$ (d-f, in cm$^{-3}$), CER (g-i, in μm) and CLWP (j-l, in g·m$^{-2}$), and accumulated precipitation (m-o, in mm) averaged over the simulation period from the Clean (left column) and Control (middle column) experiments. The right column shows the differences between the two experiments (Control minus Clean). NMB in the top-right corner of each panel represents the relative change of Control compared to Clean.

Cloud development under both pollution regimes and the enhancing effect of continental aerosols on clouds exhibit regular variations with RH and LTS (Fig. 6). Since tests shown in Fig. S7 indicate that LTS and RH do not exhibit noticeable lagged effects on clouds at 1-, 3-, and 6-hour lag times in this case, lag analysis is not applied in this study. Between the surface and 1300 m, where more than 97% of the cloud droplets in this case are located, supersaturation occurs mainly in humid environments with RH above 60% and increases with LTS. Except in a few weak clouds under relatively low RH conditions (<85%), where the reduction in droplet size due to continental aerosols intensifies evaporation and weakens the cloud, continental aerosols generally increase $N_d$ and CLWP across ECO. In the Clean experiment, high $N_d$ and CLWP occur more frequently in low-LTS environments dominated by updrafts, compared to high-LTS conditions. In contrast, the Control

experiment shows a significantly higher frequency of high $N_d$ and CLWP under high-LTS conditions, resulting from the combined effects of substantial continental aerosol transport associated with winter monsoon and strong condensation, as described in Section 3.1. Additionally, we analyze the cloud lifetime effect induced by continental aerosols based on the counts of liquid-phase cloud samples (defined as the total count of grid cells where and when CLWP exceeds 1 $g \cdot m^{-2}$ and no ice-phase particles are present). The results (Figs. 6m–o) show that under relatively moist conditions (RH > 80 %), continental aerosols extend cloud lifetime, with this effect first decreasing and then increasing with LTS. The extension is strongest when LTS is either low or high, corresponding to vigorous updrafts (invigorating frequent cloud hydrometeors) and strong cold advection (causing intense condensation and maintaining cloud), respectively. Under relatively dry conditions (RH < 80%), enhanced evaporation driven by aerosol-induced reductions in CER dominates, generally accelerating cloud dissipation and shortening cloud lifetime. However, cloud lifetime at the high-LTS tail ($\geqslant$24 K) is extended due to suppressed entrainment. As shown by the entrainment intensity (Fig. S8) under different LTS conditions in dry environments (RH < 80%)—informed by the methodology of Jia et al. (2019) but adapted for model outputs, where entrainment intensity is defined as the difference in CLWC between the layers immediately below (non-entrainment zone) and above (entrainment zone) the vertical layer with maximum CLWC in clouds with CLWP > 1 $g \cdot m^{-2}$, divided by the maximum CLWC (i.e. ($CLWC_{non-entrainment} - CLWC_{entrainment}$) / $CLWC_{max}$)—entrainment intensity in environments with LTS $\geqslant$ 24 K is clearly lower than those with LTS < 24 K, indicating suppressed entrainment under stable atmospheric conditions. Furthermore, the comparison between Control and Clean experiments in the high-LTS environment confirms that enhanced cloud development induced by continental aerosols further mitigates the weakening of clouds by entrainment in such weak-entrainment regimes.

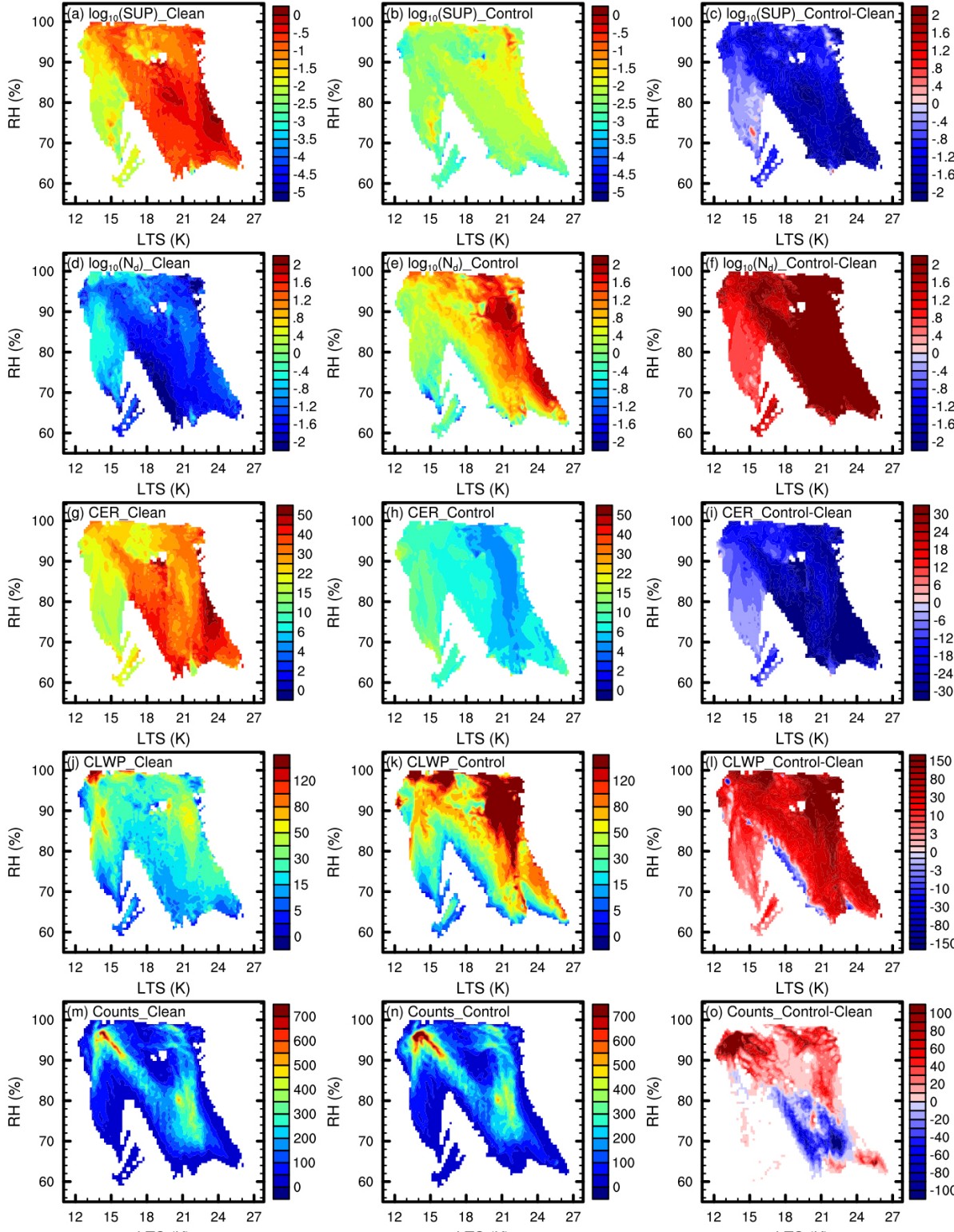

**Figure 6.** In-cloud mean supersaturation (a-c, in %) in base-10 logarithms, $N_d$ (d-f, in cm$^{-3}$) in base-10 logarithms, CER (g-i, in μm), and CLWP (j-l, in g·m$^{-2}$), as well as the counts of liquid-phase cloud samples (m-o) variations with RH and LTS, are presented for the Clean (left column) and Control (middle column) experiments, along with the difference between them (Control minus Clean, right column) during the simulation period. Supersaturation and RH are column-mean values from the surface to 1300 m. All liquid-phase cloud samples with a CLWP greater than 1 g·m$^{-2}$ and no ice-phase particles throughout the column are included. Samples are binned into 100 × 100 bins based on horizontal and vertical coordinates. To ensure that the differences between the two experiments represent the impact of continental aerosols on the same cloud processes, the binning

is performed using the meteorological fields from the Control experiment rather than the respective meteorological fields from each individual experiment. In the figure, counts represent the sum of samples within each bin, while the others represent the average of samples within each bin.

430    The variations of precipitation parameters, including $N_r$, rainwater path (RWP), and hourly precipitation, with RH and LTS are shown in Fig. 7. Similar to cloud properties, both $N_r$ and RWP increase with LTS in both experiments. Continental aerosols generally suppress precipitation by reducing droplet size, but instances of precipitation enhancement are also observed. Increases in $N_r$ and RWP mainly occur under high-humidity conditions (RH > 90%), while the LTS conditions linked to increased precipitation appear relatively irregular. The effects of continental aerosols on $N_r$ and RWP are highly consistent. Due

435    to differences in raindrop size and vertical position, the hourly precipitation generally aligns with changes in $N_r$ and RWP, but exhibits discrepancies in some details. Compared to precipitation, which has a lagged response, $N_r$ and RWP provide more immediate and accurate tracking of the aerosol-cloud-precipitation processes. Subsequent analyses of precipitation therefore focus on $N_r$ and RWP.

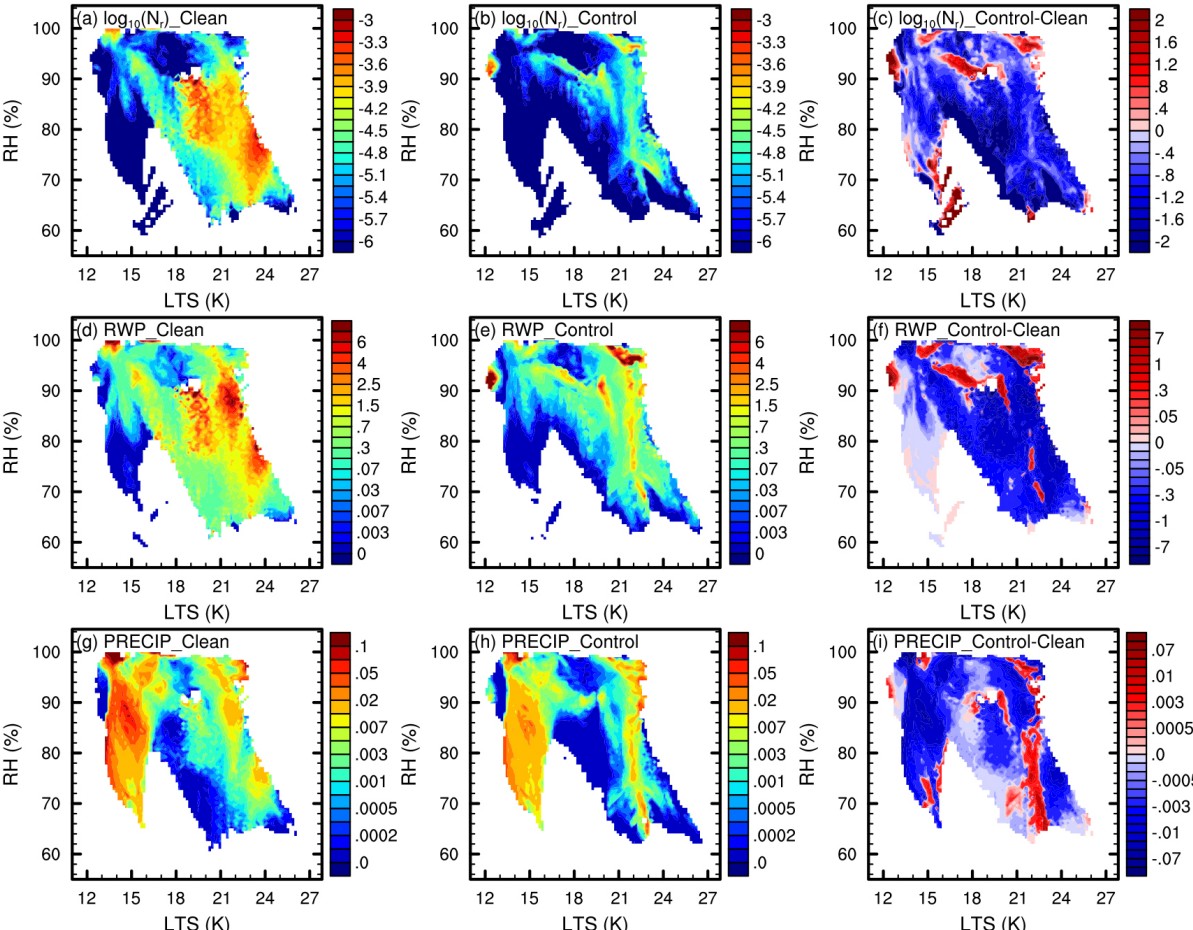

440    **Figure 7.** In-cloud mean $N_r$ (a-c, in cm$^{-3}$), RWP (d-f, in g·m$^{-2}$), and hourly precipitation (g-i, in mm·h$^{-1}$) variations with RH and LTS, are presented for the Clean (left column) and Control (middle column) experiments, along with the difference between them (Control minus Clean, right column) during the simulation period. The data processing method is the same as that used in Fig. 6.

445    We analyze samples with enhanced and reduced precipitation under the influence of continental aerosols (selecting samples with RWP increases and decreases greater than 0.001 g·m$^{-2}$) to further understand the mechanism of continental aerosol effects on precipitation. For the precipitation reduction samples (Figs. 8a-d and S9a-f), the differences in $N_r$ and RWC between the two experiments at high levels (1600-4000 m) are small. However, at the low levels (around 500 m) where the majority of cloud

droplets are located, the decrease in CER significantly suppresses precipitation, causing it to reduce by an order of magnitude. For the precipitation enhancement samples, under the influence of continental aerosols, the column-mean $N_d$ is 2.64 times higher than in the precipitation reduction samples. As shown in Figs. 8e-h and S9g-l, the enhanced vertical development of low-level clouds (below 1500 m) strengthens collision-coalescence (producing large droplets), and, coupled with a substantial increase in droplet numbers in the $N_d$-limited relatively high-level clouds (above 1500 m), collectively drives the pronounced rise in $N_r$ and RWP across all levels with liquid-phase cloud.

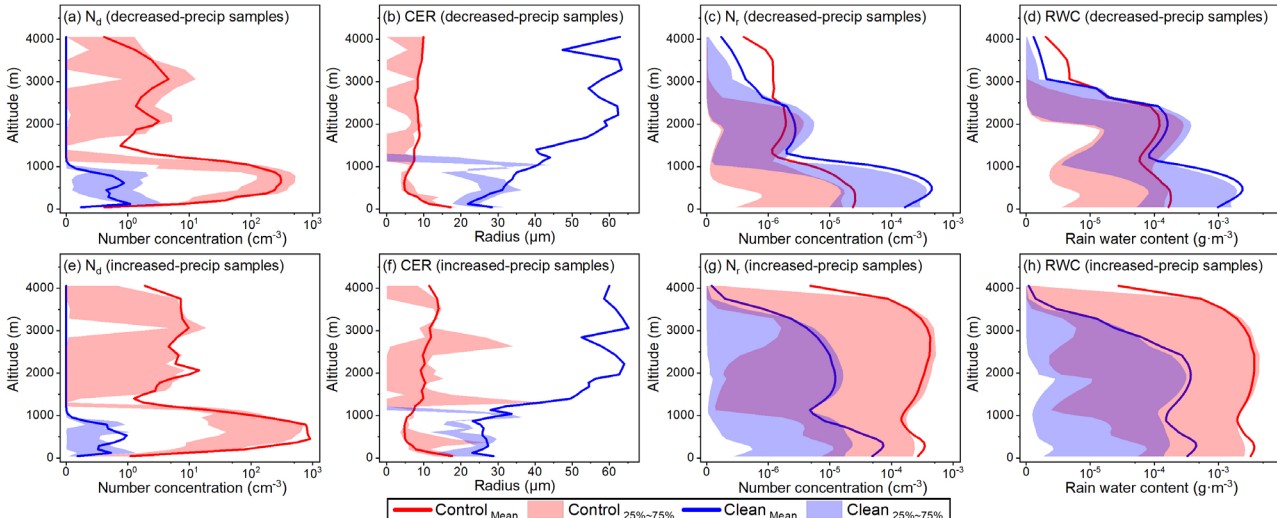

**Figure 8.** Vertical profiles of the average $N_d$ (a and e), CER (b and f), $N_r$ (c and g), and rainwater content (RWC, d and h) for precipitation reduction (a-d) and enhancement (e-h) samples under the influence of continental aerosols.

## 4.2 The role of meteorological conditions in ACI under different environments

In Section 4.1, by examining the variation of cloud properties with LTS and RH, we gain an overall understanding of how liquid-phase clouds under the two pollution regimes and the influence of continental aerosols respond to meteorological conditions. However, this apparent cloud response also contains embedded signals of ACI induced by aerosol variability within each regime. In this section, we analyze the synergistic variation of cloud properties with meteorological conditions and aerosols, focusing on three key issues to further clarify the influence of meteorological conditions on ACI: (1) What role do meteorological conditions play in ACI variations with aerosols? (2) How much impact can changes in meteorological conditions exert on ACI under given aerosol conditions? (3) How does the sensitivity of ACI to meteorological and aerosol conditions change with environmental variations?

In addition to cloud properties such as CER, CLWP, and RWP, we use ratios including $N_d/N_a$, CLWP/$N_a$, and RWP/$N_a$ to characterize activation efficiency, as well as the response rates of CLWP and RWP to $N_a$, to illustrate variations in ACI under varying meteorological and aerosol conditions. First, we examine the synergistic effects of atmospheric dynamics (represented by LTS) and aerosols, as shown in the left two columns of Fig. 9. Under the clean regime, aerosol activation efficiency shows little sensitivity to changes in LTS due to the very low aerosol number with high proportion of large particles from natural emissions (Fig. 10a), which maintains cloud processes in an aerosol-limited state and results in high activation efficiency across varying LTS and $N_a$ conditions. As $N_a$ increases, aerosols continue to activate with high efficiency to form cloud droplets, thereby promoting an increase in CLWP. However, in the aerosol-limited state, the weak role of collision-coalescence and the dominance of condensation cause the CER reduction induced by increased $N_a$ to suppress precipitation, while the growth rate of CLWP also gradually slows. The influence of LTS is more pronounced in the polluted regime, which contains a large number of aerosols dominated by small particles (Fig. 10a). As indicated by the covariation between LTS and $N_a$ in Figure 2 and the circulation changes under different LTS regimes in Figure 1, the overall increase in both LTS and $N_a$ signifies that the ECO is increasingly influenced by continental cold advection. Under low LTS, increases in $N_a$ are accompanied by a weakening of cold

advection on updrafts, whereas under high LTS, increases in $N_a$ correspond to intensified cold advection. This results in reduced activation efficiency and CLWP growth rate when $N_a$ exceeds $10^{2.8}$ (~631) cm$^{-3}$ under low LTS, while under high LTS, activation efficiency, and growth rates of CLWP and RWP all increase with $N_a$ (Column 2 of Fig. 9). Moreover, as CER decreases and CLWP increases, RWP strengthens under the polluted regime, compared to its weakening under the clean regime (Figs. 9e-f, i-j and q-r), reflecting the dominant role of collision-coalescence among numerous small cloud droplets (Fig. 10b).

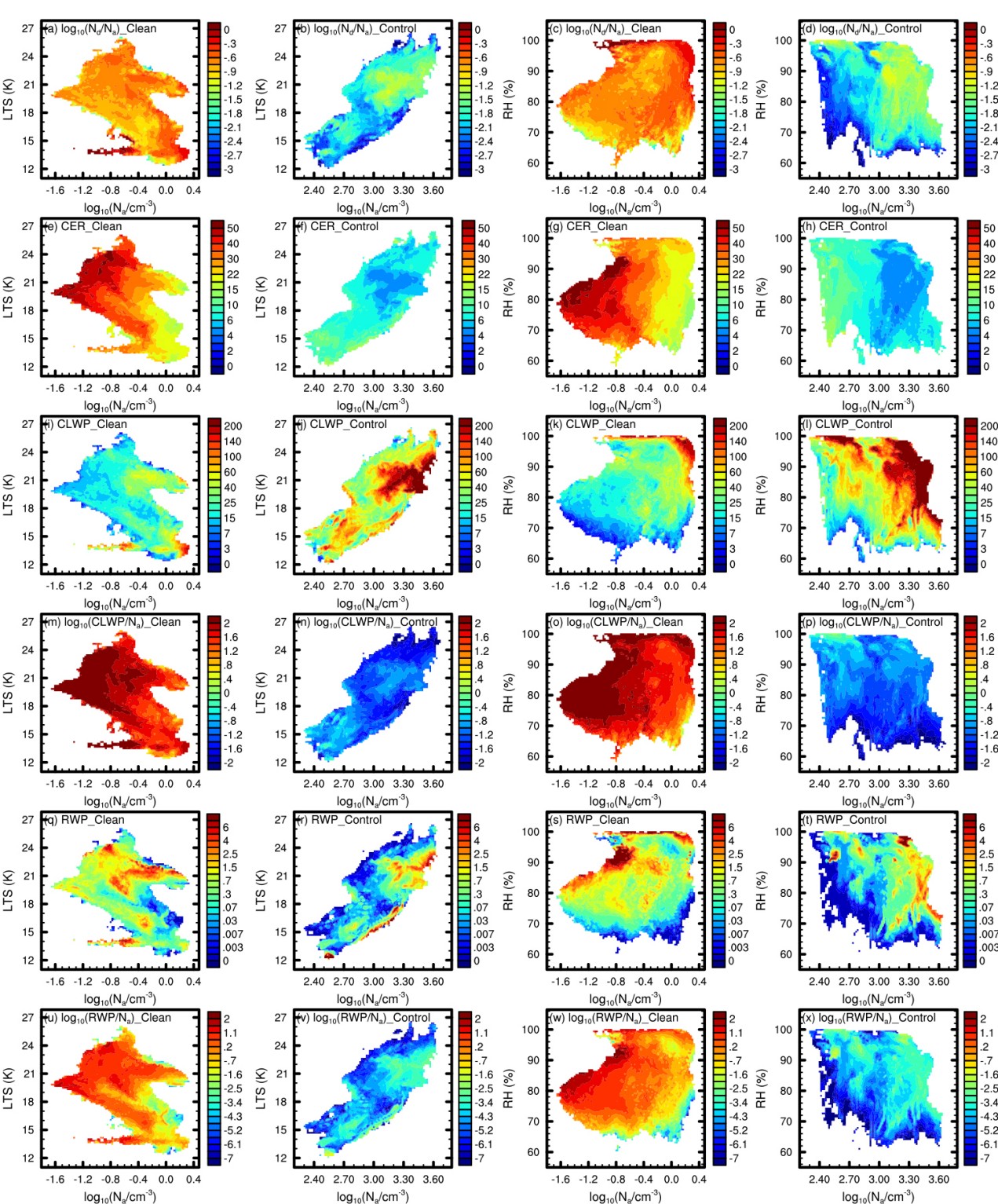

**Figure 9.** Variations of in-cloud mean $N_d/N_a$ (a-d, in cm$^{-3}$·cm$^3$) in base-10 logarithms, CER (e-h, in μm), CLWP (i-l, in g·m$^{-2}$), CLWP/$N_a$ (m-p, in g·m$^{-2}$·cm$^3$) in base-10 logarithms, RWP (q-t, in g·m$^{-2}$), and RWP/$N_a$ (u-x, in g·m$^{-2}$·cm$^3$) in base-10 logarithms

with meteorological conditions (LTS, left two columns, and RH, right two columns) and $N_a$ in base-10 logarithms for the Clean (first and third columns) and Control (second and fourth columns) experiments. Except for the use of respective meteorological elements and $N_a$ from each experiment during sample binning—to reflect changes specific to each experiment—the data processing method remains consistent with that applied in Fig. 6.

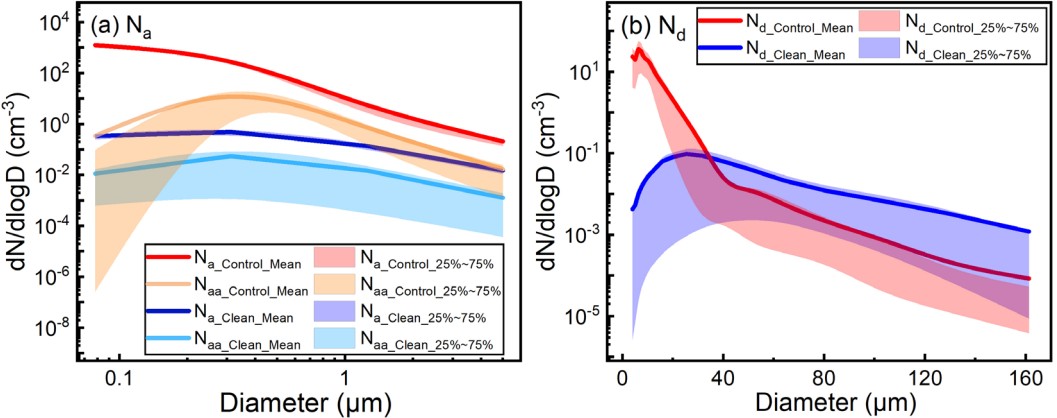

**Figure 10.** In-cloud mean aerosol (a, $N_a$ as total aerosol number concentration, $N_{aa}$ as activated aerosol number concentration) and droplet (b) size distribution in the Control and Clean experiments (In WRF-Chem-SBM, each cloud droplet corresponds to an activated aerosol particle, meaning $N_d$ and $N_{aa}$ are equivalent in total quantity. Fig. b shows the spectral distribution of cloud droplets through the variation of $N_d$ across different bins, while Fig. a displays the activation efficiency of aerosols by comparing $N_a$ and $N_{aa}$ of the four aerosol bins).

Compared to the low sensitivity of ACI to LTS under clean regime and the differential response of ACI at different LTS levels under polluted regime, the synergistic response of ACI to RH and $N_a$ exhibits more clearly regular variations (right two columns in Fig. 9). Under the clean regime, clouds show much stronger sensitivity to RH than to LTS (as indicated by the correlations presented in Table S1), with activation efficiency, CER (which, due to intensified droplet competition under high RH and high activation efficiency, is the only cloud property in Table S1 with a lower correlation to RH than to LTS), CLWP, and RWP all generally increasing with RH. Under the polluted regime, characterized by abundant aerosols and a water-vapor-limited environment, CER, $N_d/N_a$, $CLWP/N_a$, and $RWP/N_a$ all exhibit higher sensitivity to RH (on average over 46.98% higher) compared to the clean regime (Table S1). This contrasts with CLWP and RWP, which show slightly and significantly lower sensitivity to RH than clean regime, respectively, due to interference from sensitivity to $N_a$ caused by these collision-coalescence-dominated cloud properties. Overall, the coordinated variations of cloud properties with LTS/RH and $N_a$ reveal the regulatory role of meteorological fields on ACI. Specifically, the shift from convection-dominated regimes at low LTS to cold advection-dominated regimes at high LTS modulates the qualitative impact of aerosols on cloud properties (under polluted regime), while the level of RH governs the quantitative effect of aerosols on cloud development.

The variation of ACI with RH and LTS demonstrates the influence of thermodynamic meteorological conditions. However, due to the delayed response of clouds to changes in thermodynamic conditions, the variation of ACI with RH/LTS exhibits some discontinuities and uncertainties. We further investigate the sensitivity of ACI to meteorological conditions based on supersaturation (column-mean from the surface to 1300 m), which serves as a more immediate and responsive indicator (Fig. 11). Cloud properties in both regimes exhibit distinct regular variations with supersaturation. Under the clean regime, $N_d$, CER, CLWP, and RWP all increase with supersaturation. Under this aerosol-limited regime, cloud properties are also sensitive to aerosols. As $N_a$ increases, $N_d$, CLWP, and RWP generally show consistent increases, while CER correspondingly decreases. A notable deviation from this overall trend occurs when $N_a$ falls within the range of $10^{-0.9}$ (~0.13) to $10^{-0.4}$ (~0.40) cm$^{-3}$ and supersaturation is below 0.1%, where RWP shows a relatively high-value band. This is because, within this $N_a$ range, $N_d$ increases with $N_a$, while CER remains relatively high (approaching 40 μm even under low supersaturation). When $N_a$ exceeds $10^{-0.4}$ (~0.40) cm$^{-3}$, further increases in $N_a$ under low supersaturation cause CER to rapidly decrease to around 15 μm, thus

suppressing precipitation, while high RWP appears only under high supersaturation.

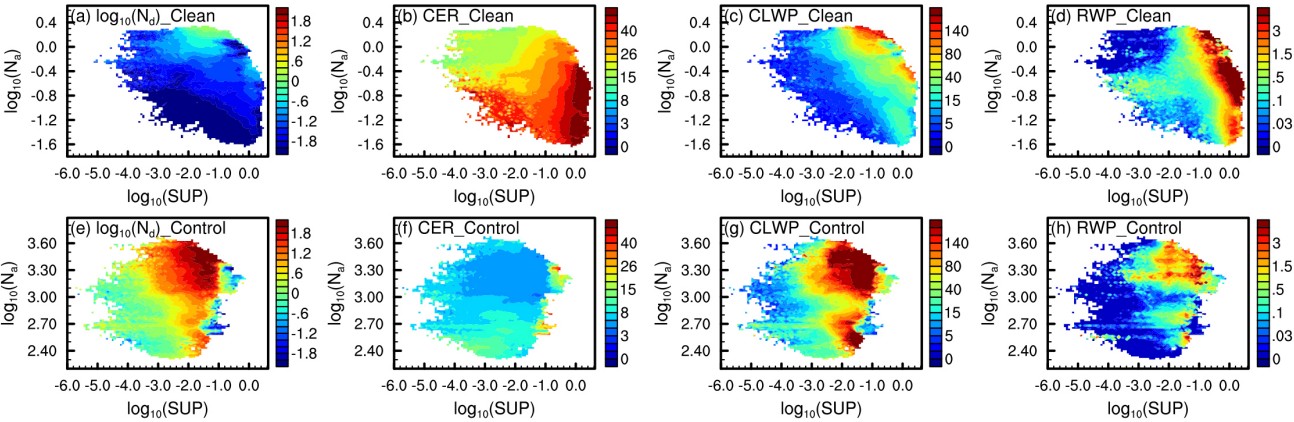

**Figure 11.** Variation of in-cloud mean $N_d$ (a and e, in $cm^{-3}$) in base-10 logarithms, CER (b and f, in μm), CLWP (c and g, in $g \cdot m^{-2}$), and RWP (d and h, in $g \cdot m^{-2}$) with $N_a$ (in $cm^{-3}$) and supersaturation (SUP, in %) in base-10 logarithms for the Clean (a-d) and Control (e-h) experiments during the simulation period. The data processing method is the same as that used in Fig. 9.

In clear contrast to the clean regime, $N_d$, CLWP, and RWP under the polluted regime exhibit non-monotonic responses to both supersaturation and $N_a$: they initially increase and then decrease with supersaturation, and show a decrease followed by an increase with $N_a$. At relatively low to moderate supersaturation levels ($<10^{-0.9}$ %, i.e., $<\sim0.13$ %), increased supersaturation promotes aerosol activation, resulting in elevated $N_d$, CLWP, and RWP. However, as supersaturation continues to rise, many droplets grow to moderate size. Their stronger competition for water vapor relative to small droplets, along with the rapid evaporation of the latter, leads to a significant reduction in the number of small droplets. This weakens collision-coalescence and ultimately inhibits the formation of large droplets. At relatively high supersaturation ($>10^{-0.8}$ %, i.e., $>\sim0.16$ %, due to the substantial depletion of supersaturation by aerosols, although this supersaturation is not high in absolute terms, it represents a relatively high value among all Control samples), although CER increases, only moderate-sized droplets increase, while both raindrop and small droplets decrease significantly compared to relatively moderate ($10^{-2.0}$ to $10^{-0.9}$ %, i.e., $\sim0.01$ to 0.13 %) supersaturation (as shown in Fig. 12), resulting in reductions in $N_d$, CLWP, and RWP. The non-monotonic responses of cloud properties to supersaturation primarily reflect microphysical influences, while the non-monotonic response to $N_a$ arises from the coordinated variations between $N_a$ and meteorological conditions. As shown in Figs. 1 and 2, under winter monsoon conditions, continental cold air advection from the northwest leads to high LTS generally coinciding with high $N_a$ over ECO, while warm, moist, and clean air flows from the south result in low LTS typically associated with low $N_a$. High $N_a$ ($>10^3$ $cm^{-3}$) and low $N_a$ ($<10^{2.8}$, $\sim630$ $cm^{-3}$) generally correspond to strong cold advection under high LTS and robust updrafts under low LTS, respectively, both promoting elevated CLWP and higher $N_d$ compared to moderate $N_a$ levels ($10^{2.8}$ to $10^3$ $cm^{-3}$). In contrast, moderate $N_a$ environments experience neither sufficiently strong cold advection nor sustained updrafts (as the average variations of temperature advection and vertical wind speed with $N_a$ in ECO, shown in Fig. S10), and these opposing mechanisms often counteract each other, leading to reduced $N_d$, CLWP, and RWP. The significant impact of the synchronous intensification of cold advection and $N_a$ on ACI is only evident in the control experiment corresponding to the polluted regime. In the clean experiments without continental aerosols, the variation in $N_a$ shows no clear correlation with cold advection. Moreover, due to minimal precipitation during the study period (maximum and mean hourly precipitation for all liquid-phase cloud samples in the two experiments are 0.25 and 0.004 $mm \cdot h^{-1}$, respectively), precipitation closely follows changes in CLWP and does not significantly feedback on cloud processes.

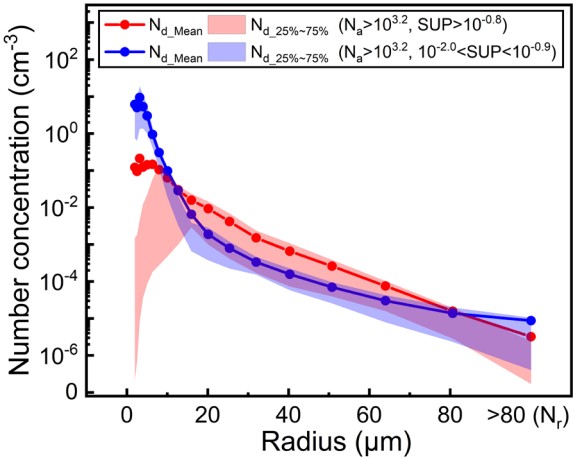

**Figure 12.** Column-mean cloud droplet and raindrop (droplet with a radius greater than 80 µm) spectral distributions (mean and 25th to 75th percentile range) in relatively moderate ($10^{-2.0}$ to $10^{-0.9}$ %, i.e., ~0.01 to 0.13 %, blue line) and relatively high supersaturation (>$10^{-0.8}$ %, i.e., >~0.16 %, red line) environments under high $N_a$ (>$10^{3.2}$ cm$^{-3}$, i.e., >~1585 cm$^{-3}$) during the simulation period.

## 5 Summary

Large uncertainty remains in ACI due to its complex physical and dynamical mechanisms. Many studies have been conducted to examine the mechanisms of ACI and have found that the aerosol-cloud relationship varies significantly under different meteorological conditions. In this study, WRF-Chem-SBM model, which couples spectral bin cloud microphysics with an online aerosol module, is used to simulate a wintertime liquid-phase cloud case over ECO. The model's high-precision simulation of aerosol-cloud processes, the case featuring rich variability in meteorological, aerosol, and cloud properties, the comparison between the Control experiment (representing a polluted regime with both continental and marine aerosols) and the Clean experiment (representing a clean regime with only marine aerosols), and the isolation of radiative effect interference through model configuration collectively support an in-depth understanding of the sensitivity of ACI to meteorological conditions.

To ensure the reliability of the simulation results and quantify the impact of four-dimensional data assimilation, we evaluate the simulations using multi-source observational data, including meteorological fields, aerosols, clouds, and precipitation. The evaluation indicates that data assimilation generally improves meteorological simulations, with particularly notable improvements in dewpoint depression (with the average correlation between simulated and observed dewpoint depression increasing by 16%). This effectively enhances the model's ability to simulate ACI. With support of assimilation, the model (Control experiment) well reproduces the values and distributions of meteorological fields, aerosols, and cloud properties, with correlation coefficients above 0.72 and errors within ±13.3% (apart from systematic errors in cloud top parameters) relative to observations. The model reasonably reproduces precipitation amounts and spatial distribution of precipitation bands. However, due to slight positional discrepancies between simulated and observed rainbands, the agreement (r = 0.57, NMB = –27.3%) between simulation and observation is lower than for other parameters. Evaluation of the spatial distribution and uncertainties in simulations and observations indicates that the model reproduces meteorological fields, aerosols, clouds, and precipitation within acceptable error ranges, thereby supporting the reliability of the results. The consistency of meteorological fields between the Control and Clean experiments, along with the inter-experiment differences in ACI, further demonstrate the robustness of the simulated signals.

The simulation results reveal the substantial impact of continental aerosols on liquid-phase clouds over ECO, leading to a 248-fold increase in $N_d$, a 77.74% decrease in CER, and a 2.75-fold rise in CLWP on average. This impact and cloud processes under polluted regime are enhanced with increasing LTS, which highlights the dominant role of cold advection—which co-varies with $N_a$—over updraft in the context of the winter monsoon. Continental aerosols extend cloud lifetime (actually cloud

sample counts in this study) in moist environments, with this enhancement strongest under low and high LTS conditions, corresponding to vigorous updrafts (invigorating frequent cloud hydrometeors) and strong cold advection (causing intense condensation and maintaining cloud), respectively. In dry environments, the reduced CER and intensified evaporation caused by continental aerosols generally shorten cloud lifetime. However, in high LTS environments (LTS $\geqslant$24 K) with relatively weak entrainment, the overall enhancement of cloud development due to continental aerosols further offsets the weakening effect of entrainment, resulting in increased cloud lifetime. Moreover, continental aerosols exhibit both enhancing and suppressing effects on precipitation over ECO, but these effects do not show clear sensitivity to RH or LTS. Instead, the difference between enhanced and suppressed precipitation samples lies primarily in the intensity of cloud processes, with the former exhibiting an average $N_d$ 2.64 times higher than the latter. In weak cloud processes, aerosol-induced increases in $N_d$ and reductions in CER significantly suppress precipitation, whereas in intense cloud processes, enhanced vertical development and vigorous collision-coalescence among numerous droplets in low-level clouds, and a substantial increase in droplet numbers in the $N_d$-limited high-level clouds, both lead to increased precipitation.

Building on the overall analysis of cloud process variations with LTS and RH, we further explore the sensitivity of ACI to meteorological conditions by analyzing cloud responses to meteorological factors at varying $N_a$ levels under different regimes. The variation with LTS reveals the sensitivity of ACI to meteorological conditions from a dynamical perspective. In the aerosol-limited clean regime, aerosol activation efficiency shows no clear sensitivity to LTS. As $N_a$ increases, aerosols still activate efficiently to form cloud droplets, leading to an increase in CLWP. However, under this aerosol-limited condition, the weak role of collision-coalescence and the dominance of condensation suppress precipitation as CER decreases, while the growth rate of CLWP also slows with the reduction of CER. Under the polluted regime, characterized by abundant aerosols, cloud properties exhibit strong sensitivity to LTS and show contrasting responses to $N_a$ under different dynamical conditions. Furthermore, unlike the clean regime where RWP decreases with CLWP, under the polluted regime RWP increases with CLWP, indicating the dominant role of collision-coalescence under this regime. Cloud processes under both clean and polluted regimes show clear sensitivity to RH, with the water-vapor-limited polluted regime exhibiting higher sensitivity.

We further investigate the sensitivity of ACI to meteorological conditions based on supersaturation, which serves as a more immediate and responsive indicator. Clouds in both regimes exhibit high sensitivity to supersaturation. Under the clean regime, $N_d$, CER, CLWP, and RWP all increase with supersaturation, and cloud properties are also sensitive to aerosols due to the aerosol-limited state. Under the polluted regime, the initial decrease followed by an increase in cloud properties with $N_a$, and their increase followed by a decrease with supersaturation, respectively reflect the influences of atmospheric dynamical conditions and collision-coalescence processes.

In this study, building upon existing ACI theory, we analyze the sensitivity of ACI to meteorological conditions through high-resolution modeling of a case characterized by a small spatiotemporal scale yet rich in meteorological-aerosol-cloud variations. Numerous observational and modeling studies have investigated the mechanisms and impacts of ACI (Saleeby et al., 2010; Bennartz et al., 2011; Ma et al., 2018; Jia et al., 2019a, 2022; Guo et al., 2022; Haghighatnasab et al., 2022), as well as their dependence on meteorological conditions (Salma et al., 2021; Zheng et al., 2022; Liu et al., 2024) and aerosol properties (Reutter et al., 2009; Hudson and Noble, 2014). While these efforts have substantially advanced our understanding of ACI, several limitations remain. For instance, (1) bulk microphysical parameterizations may distort ACI signals, quantitatively and even qualitatively; (2) quantitative and qualitative conclusions derived from case studies may not hold under different conditions; and (3) analyses of ACI sensitivity to individual factors are often confounded by co-varying influences, leading to uncertainties in signal separation. This study addresses these issues by: (1) use of high-resolution spectral bin microphysics modeling to better represent realistic ACI processes; (2) selection of a case exhibiting significant co-variations in meteorology, aerosols, and clouds to support a relatively comprehensive analysis; and (3) examination of ACI responses to the co-variation of different meteorological factors and of meteorological factors and aerosols, enhancement of ACI signal detectability through large $N_a$ differences between Control and Clean experiments, and improvement of meteorological consistency and signal robustness via four-dimensional data assimilation and the disabling of ARI. This study helps to clarify the mechanisms behind the nonlinear variation of ACI with environmental conditions and reduces the associated uncertainties.

***Code availability.*** The WRF-Chem model code can be downloaded from https://www2.mmm.ucar.edu/wrf/users/download/get_sources.html (University Corporation for Atmospheric Research, 2024). The WRF-Chem-SBM model code can be obtained by contacting Dr. Jiwen Fan (fanj@anl.gov) of Argonne National Laboratory.

***Data availability.*** The namelist file and output of the model can be downloaded from https://doi.org/10.5281/zenodo.17219355 (Zhao, 2025). NCEP data sets (https://doi.org/10.5065/39C5-Z211, Satellite Services Division et al., 2004; https://doi.org/10.5065/4F4P-E398, NCEP et al., 2004; https://doi.org/10.5065/D65Q4T4Z, NCEP et al., 2015), CAM-chem model output (https://doi.org/10.5065/NMP7-EP60, Buchholz et al., 2019), MICAPS meteorological fields (http://www.nmc.cn, National Meteorological Centre of China, 2009), near-surface $PM_{2.5}$ observations (https://air.cnemc.cn:18007, China National Environmental Monitoring Center, 2023), IMERG precipitation (https://doi.org/10.5067/GPM/IMERGDF/DAY/06, Huffman et al., 2019), and MODIS aerosol (https://doi.org/10.5067/MODIS/MOD04_L2.061, Levy et al., 2015) and cloud (https://doi.org/10.5067/MODIS/ MOD06_L2.061, Platnick et al., 2015) data can be accessed from the corresponding websites or references.

***Author contributions.*** JZ and XM designed and conducted the model experiments, analyzed the results, and wrote the paper. XM developed the project idea and supervised the project. XM and JQ proposed scientific suggestions and revised the paper. TY collected and processed the data.

***Competing interests.*** One of the (co-)authors is a member of the editorial board of Atmospheric Chemistry and Physics, and the authors have no other competing interests to declare.

***Acknowledgements.*** The numerical calculationsin this paper were conducted in the High-Performance Computing Center of Nanjing University of Information Science & Technology. We express our gratitude to Dr. Jiwen Fan of Argonne National Laboratory for providing the code for the WRF-Chem-SBM model. We are grateful to the National Aeronautics and Space Administration, the National Centers for Environmental Prediction, the MEIC support team, the Chinese National Meteorological Center, and the China National Environmental Monitoring Center for providing the MODIS and GPM data, FNL and observation subsets, MEIC emission inventory, MICAPS data, and $PM_{2.5}$ data, respectively.

***Financial support.*** This research has been supported by the National Natural Science Foundation of China (grant nos. 42575082, 42061134009 and 41975002), the Second Tibetan Plateau Scientific Expedition and Research program (grant no. 2019QZKK0103), the China Scholarship Council program (grant no. 202309040034), and the Postgraduate Research and Practice Innovation Program of Jiangsu Province (grant no. KYCX22_1151).

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
