# Peer review of "Figure S1. Vertical profiles of average temperature (a and f), water vapor content (b and g), absolute value of vertical wind speed (c and h), vertical wind speed (d and i), and supersaturation (e and j) over ECO for the Clean (a-e) and Control (f-j) experiments."

_EGUsphere, 2025_

## Referee Comment (RC1)

**Review of 'How meteorological conditions influence aerosol-cloud interactions under different pollution regimes'**

This study uses WRF-Chem to simulate marine liquid clouds near the Eastern China Ocean during a selected winter period. The experiments are designed to examine aerosol effects on cloud and precipitation by comparing clean versus polluted scenarios. Results show that aerosols extend cloud lifetime in moist conditions but shorten it in dry conditions. In the polluted regime, continental aerosols lead to higher Nd and CLWP and smaller CER, while precipitation responses are mixed overall. Under the clean regime, aerosol activation is efficient and responses are clearer with supersaturation. In polluted conditions, clouds exhibit mixed responses tied to different cloud processes, reflecting regime-dependent ACI sensitivity.

While the overall narrative and logic read fine, I found multiple occasions where the interpretation and discussion are rather vague or misleading. As such, I have questions and comments on technical and interpretive details. The comments below include several general comments and more specific comments, laid out in the narrative sequence when reading the manuscript.

**General Comments**

In the Introduction, please situate this work within prior LES efforts (WRF-SBM, WRF-Chem, SAM, etc.) over the East China Ocean and synthesize what they concluded about ACI. Then clarify the remaining gap and why a chemistry-aware framework is needed here: specifically, what does WRF-Chem capture that a physics-only perturbation of aerosol number in WRF/WRF-SBM cannot? Because much of your analysis emphasizes dynamical/thermodynamic linkages, please consider discussing how chemical pathways modify those linkages and how your design advances beyond prior 'simple number-perturbation' studies.

Subsection 3.2, 'Experiment setup', should be moved into the WRF-Chem description or at least earlier in Section 3, preferably before Figure 2.

If I am understanding correctly, the middle columns in both Figs. 4 and 5 are from the Control experiment. If so, please clarify why the spatial distributions for Nd and CLWP in Fig. 5 differ noticeably from those in Fig. 4 (e.g., an empty area in Fig. 4 but not in Fig. 5, etc.), and why precipitation also exhibits subtle differences. Please ensure consistent sampling/calculation for region maps like these.

When describing figure results, please also list a plain-number expression alongside the scientific expression to help intuitive interpretation. For instance, at L402: 'At sub-moderate supersaturation levels (…)' include the normal expressed percentage values ($<10^{-0.9}\%$ → $<$ ~0.12%).

Please carefully revisit whether the cited supporting figures actually show and support the associated statements. Some cases are included in detail comments below.

**Specific Comments**

**L47.** You may want to narrow the statement down to 'satellite observation…' given the reference cited.

**L62.** Please include a brief description of the following sessions of the manuscript.

**L96.** What is the uncertainties of this Nd retrieval compared with the in-situ observation.

**L99.** Please specify the (approximate) local hour of Terra overpasses over ECO.

**L111.** Please specify the matching logic (e.g., whether the WRF output at the nearest time step or an average over surrounding times was used). For a given WRF grid cell, approximately how many MODIS L2 records were averaged?

**L126.** How much uncertainty will these two different modeled Nd derivations introduce into the later ACI analysis? Can you show a comparison of calculated/simulated Nd vs. directly output Nd?

**L141.** Please specify the lat/lon range of the red box.

**L152.** Is there a specific reason for selecting 1300 m? Also, how does the model cloud-top height compare with the satellite CTH?

**L152.** Please correct me if I'm wrong, but by eye are LTS and RH exactly the same between the Control and Clean regimes, or are the differences too subtle to visualize?

**L154.** Do the variables shown in Fig. 2 represent domain means or cloud-sample means?

**L155.** This 'dominant-factor transition' statement needs physical-mechanism support. Otherwise, omit it and reframe as a description of wind direction (as in the next sentence), or simply state that the domain was impacted by cold-air advection from the northwest.

**L162.** …inhibiting updraft (Fig. S1i)

**L163.** I wouldn't say 'rapid' given the wind speed isn't significantly faster than in the LTS < 17 scenario.

**L171.** 'low-pollution' might be misleading, try 'moderate' or 'relatively low', given that 'clean' is used later.

**L172.** That is an ultraclean condition; I would like to see the corresponding Nd.

**L173.** Which case? Are you referring to the polluted or clean case as reflecting typical variations in the joint field over your study region? Please clarify. I recommend retuning this paragraph so the message is tied more tightly to this particular study if you want to keep these general statements.

**L211.** It looks like the Control has larger error bars (and slightly more outliers) than Control_NoDA, especially for the wind components. I recognize the better correlation in Fig. S2; however, can you quantify mean bias or RMSE between these two and observations to support your statement?

**L240.** Please also note in the caption that the lower subpanels are for the red-box domain.

**L297.** If you only present counts in Fig. 6m-o, I would not call it 'occurrence frequency', which usually meant for depict fraction or percentage. Suggest just use 'counts' or 'samples'. Or maybe you can get the actual fractional frequency of occurrence of clouds.

**L301.** Do you have a physical explanation for the increased cloud lifetime at the high-LTS tail (>24 K), which also corresponds to the lowest RH? I wonder if entrainment mixing is inhibited, given that Nd (CER) is comparatively lower (larger) at that tail.

**L335.** It will not be intuitive to say 'more large droplets,' since the 75th percentile of CER under Control is still largely smaller than under Clean. Consider: 'collision–coalescence at lower levels can produce large droplets.'

**L336.** Have you checked cloud fraction and sample size vertically? In the Increase_ scenario, the ~3 km samples appear fewer (narrower spread), so precipitation enhancement may stem from different cloud layers (and not only from 'vertical development of the cloud layer'). If so, please make this clear.

It would help to examine vertical cross-sections of clouds (Nd, CER, Nr, etc.) for both scenarios to confirm. Since Fig. 4 suggests regional clustering for Increase_/Decrease_ samples, a SW–NE slice could work, please consider this.

**Figure 8.** The legends (e.g., Increase/Decrease_CER) are misleading. Consider renaming to clearer identifiers, e.g., CER (increased-precip samples).

**L364.** The correlation is significant but moderate; what message is this intended to convey? Because you did not show the relative vertical position of 950 hPa cold-air advection versus the cloud layers, the physical interplay is not determinate. I guess you can remove this...

**L365.** Can you remind me where did you show the Na relationship to LTS? Are you referring to Fig. 9a? And please add the support figure indentifer to this two statements.

**Figure 10.** Please clarify the units in panel (a). The in-cloud Na magnitudes in both experiments seem to far exceed the number concentrations shown previously (e.g., Figs. 2, 5). If you are showing an aerosol size distribution, the y-axis should be dN/dlogDp (cm$^{-3}$).

Also, what is the relationship between activated aerosol Naa and cloud-droplet Nd in the model outputs? Please elaborate.

**L369.** Fig. 10b provides limited support for this statement (no CLWP/RLWP shown). Please elaborate and/or provide a clear reference.

**L377.** At least CER and RWP do not increase steadily with RH. Please provide statistical support for 'stronger sensitivity,' e.g., correlations or regression slopes of cloud properties vs. LTS/RH.

**L379.** Please elaborate. If activation ratio, CLWP, and RWP largely increase with Na across RH ranges, how is 'greater sensitivity' exhibited?

**L380.** Regarding the 'manner' of cloud vs. Na being similar to high LTS (I presume Fig. 9b, 9f, etc.): would that mean cloud properties are largely impacted by aerosol loading rather than environmental parameters? If so, what 'dominated role' is cold-air advection playing? You have not established a strong physical relationship between cold advection and cloud (a correlation between column-mean Na and 950 hPa temperature is not enough). If you want to discuss the physical role of cold advection, at least show the vertical dependence of cloud properties on it. Please elaborate and provide figure support.

**L392.** The RWP sharply decrease with Na ($> 10^{-0.4}$) under low supersaturation, which reflects the precip suppression effect of aerosol, please spell it out.

**L406.** ~0.16% supersaturation is not necessarily 'high' condition.

**L407.** ~0.01% supersaturation is fairly low, again, describing the supersature with a normal expression of number will give the reader more sense.

**L410.** As above, please further explain the physical basis for cold-advection dominance. This statement is somewhat misleading: updraft strength has limited direct relationship to aerosol concentrations. Please specify which cloud process is dominated by updraft under low-aerosol conditions and how that dominance is dampened under high-aerosol conditions. In which experiment would cold-advection effects be more effective?

**L411.** Provide statistical evidence or figure support for "At moderate Na, both updrafts and cold advection are weak…"

**L449.** You may wish to point out that the extensions of cloud lifetime (actually cloud frequency in this study; Fig. 6) at low and high LTS arise from different physical reasons: low LTS/vigorous updrafts invigorate frequent cloud hydrometeors, whereas high LTS/stronger subsidence maintains the cloud layer.

**L482.** I would like to see a revised Conclusions section after you address the main-text comments.

---

## Author Comment (AC1)

**Response to the Comments of Referees**

**Journal:** Atmospheric Chemistry and Physics **Manuscript Number:** egusphere-2025-2555

Title: How meteorological conditions influence aerosol-cloud interactions under different pollution

regimes

Author(s): Jianqi Zhao, Xiaoyan Ma, Johannes Quaas, and Tong Yang

We thank the reviewers and the editor for providing helpful comments to improve the manuscript. We have revised the manuscript according to the comments and suggestions of the referees.

The referee's comments are reproduced (black) along with our replies (blue). All the authors have read the revised manuscript and agreed with the submission in its revised form.

**Anonymous Referee #1**

Using WRF-Chem-SBM, the authors conduct a realistic short-term regional simulation and a counterfactual one (same as the realistic one but without continental aerosols). They briefly showcase the model is valid by comparing the control (realistic) simulation with some observations. Then they use the pair of simulations conducted to study ACI. Ultimately, the results are mixed and show that ACI is complex and regime-dependent. I think this is a fine manuscript overall: it is interesting, relevant, and seems logical. I do have some concerns before publication (which I will ultimately support) — for the recored, I concur with Reviewer 1's concerns, though I tried not to repeat them.

We sincerely appreciate your encouraging and constructive comments and suggestions, which have greatly helped us improve the quality of this study. All of your comments have been carefully addressed and the corresponding revisions have been incorporated into the manuscript. Our detailed point-by-point responses are provided below (in blue).

**General comments:**

An interesting detail in the experiment setup is disabling the radiative effects of aerosols and clouds. That's smart, but how much does it really impact the simulations here? Could this decision be contextualized with prior studies and potentially by showing results where these effects are not disabled? How about other confounding effects — of course, you didn't choose to modify other processes (the entropy of the system is different in the simulations). I would like the authors to discuss this decision a little more, and maybe opine on what else could be done to get rid of confounding factors/feedbacks

This approach (disabling the radiative effects of aerosols and clouds) has been commonly used for separating ACI/ARI signals in modeling studies. For example, Liu et al. (2020) isolated the significant enhancement effect of ACI on the extreme heavy rainfall event in Guangzhou in 2023 by turning ARI on/off in simulations. Wang et al. (2020) used similar ARI on/off sensitivity experiments to analyze the impact of dust aerosols on radiative energy budget during a dust storm

and examined their role in the accumulation of primary pollutants over land and the formation of secondary pollutants over the ocean. Zhao et al. (2025), by comparing sensitivity experiments with ACI/ARI turned off against a baseline simulation, disentangled the contributions and mechanisms of both ACI and ARI during a complex pollution episode...

In order to examine how this approach (disabling the radiative effects of aerosols and clouds) impact the simulations, we have conducted the simulations tests (all tests use settings consistent with the Control experiment except for radiation settings) in which cloud radiative effects (Rad CLD), aerosol radiative effects (Rad AER), and both aerosol and cloud (Rad ALL), are turned on. The ECO regional average results are shown in Fig. R1. The radiative effects of aerosols and clouds show minimal influence on the temporal trends of meteorology, aerosols, and clouds (Figs. R1a-e). It is acknowledged that during specific periods, the radiative effects might alter Na through influencing transport and deposition processes and modify cloud properties by affecting meteorological and aerosol conditions. Their impact on the regional averages of Na, Nd, and CLWP exceeds 20% at certain times. In terms of ACI signals (Figs. R1f-h), the cloud radiative effect accelerates the increase of Nd with Na and the increase of CLWP with RH. In contrast, the aerosol radiative effect opposes the radiative effect of clouds. The mutual offset between these effects results in the Rad ALL test exhibiting the closest agreement with the Control experiment for these two trends. Compared to these, the relationship between clouds and LTS exhibits weaker regularity. The perturbation from ARI further increases the uncertainty, posing a significant challenge to elucidating the influence of LTS on ACI. Overall, from the perspective of ACI analysis, the quantitative deviations and enhanced qualitative uncertainty caused by ARI necessitate turning off ARI in ACI simulations. We added this description to Section 3.3 of the manuscript.

**Figure R1.** LTS (a), RH (b), Na (c), Nd (d), CLWP (e), Nd versus Na (f), as well as CLWP versus LTS (g) and RH (h) for the Control experiment and each radiative sensitivity experiment (Rad\_CLD: cloud radiative effects enabled; Rad\_AER: aerosol radiative effects enabled; Rad\_ALL: both cloud and aerosol radiative effects enabled), showing data averaged over ECO (RH is the column mean from the surface to 1300 m, and Nd and Na are in-cloud means).

This study kind of left me hopeless about the state of ACI research... While I was reading, I was hoping for something in the conclusion section to offer guidance for future research and/or deeper reflection on what all this means for the community at large. I only found something about higher resolution and bigger domains in the final paragraph, but it seems that's not going to really help, right? Is this just a difficult problem that we are not going to solve well enough?

We regret bringing this feeling to you and are sorry for the misleading statement. In the revised manuscript we have included more discussions and hope to make this clearer. Higher resolution and bigger domains might help, but our point in this manuscript is that to call more attention from the community to assess ACI under different meteorological conditions, our goal is to understand the sensitivity of ACI to meteorology under different conditions and to analyze how ACI varies with aerosols from a meteorology-centered perspective.

Numerous observational and modeling studies have investigated the mechanisms and impacts of

ACI (Saleeby et al., 2010; Bennartz et al., 2011; Ma et al., 2018; Jia et al., 2019, 2022; Guo et al., 2022; Haghighatnasab et al., 2022), as well as their dependence on meteorological conditions (Salma et al., 2021; Zheng et al., 2022; Liu et al., 2024) and aerosol properties (Reutter et al., 2009; Hudson and Noble, 2014). While these efforts have substantially advanced our understanding of ACI, several limitations remain. For instance, (1) bulk microphysical parameterizations may distort ACI signals, quantitatively and even qualitatively; (2) quantitative and qualitative conclusions derived from case studies may not hold under different conditions; and (3) analyses of ACI sensitivity to individual factors are often confounded by co-varying influences, leading to uncertainties in signal separation.

This study addresses these issues by: (1) use of high-resolution spectral bin microphysics modeling to better represent realistic ACI processes; (2) selection of a case exhibiting significant co-variations in meteorology, aerosols, and clouds to support a relatively comprehensive analysis; and (3) examination of ACI responses to the co-variation of different meteorological factors and of meteorological factors and aerosols, enhancement of ACI signal detectability through large Na differences between Control and Clean experiments, and improvement of meteorological consistency and signal robustness via four-dimensional data assimilation and the disabling of ARI. This study helps to clarify the mechanisms behind the nonlinear variation of ACI with environmental conditions and reduces the associated uncertainties.

Regarding SBM specifically, are you not willing to release the code? Has the code not been released before? Beyond the code, I invite the authors to reflect on whether SBM is the right tool here (this is similar to Reviewers 1's first general comment, but specifically about SBM). It is not readily clear to me if using SBM is better or worse than using a bulk scheme (MG2, P3, etc.)

SBM is currently incorporated in the WRF model but remains uncoupled with online aerosols, and WRF-Chem's online aerosol—cloud module only supports bulk microphysics schemes. Dr. Jiwen Fan's team has developed a coupled version that integrates SBM with online aerosols, which is referred to as the WRF-Chem-SBM model. Researchers should contact Dr. Jiwen Fan for the code.

The differences between bin and bulk treatments of cloud microphysics lead to large discrepancies in their simulations of ACI under the same dynamic and thermodynamic conditions (Fan et al., 2016). Both methods have their own advantages and limitations. The main advantage of bulk is its low computational cost, which makes it suitable for large domains and long-term simulations. However, due to (1) difficulties in accurately handling the CCN budget (Fan et al., 2012), (2) reduced sensitivity to aerosols caused by the adoption of the saturation adjustment approach, and (3) simplified treatment of the conversion from cloud water to rainwater and of hydrometeor fall velocities, bulk cannot guarantee the accuracy of ACI signals (Khain et al., 2015; Fan et al., 2016). The bin method also has some limitations, such as (1) being computationally expensive, usually applicable only to relatively small domains for short time periods, and (2) its accuracy being constrained by our theoretical understanding of cloud microphysics. Nevertheless, bin is

physically more realistic, and relevant evaluations have shown that bin outperforms bulk in reproducing cloud–rain structures and resolving ACI (Khain et al., 2015). This is also the main reason why we chose SBM. Additionally, our subsequent study will step forward on the effects of aerosol size distribution and chemical composition on ACI under different meteorological conditions, the bin-based treatment of CCN size and composition in SBM would be essential. We have added this clarification in the third paragraph of the Introduction.

**Comments I wrote while reading the manuscript:**

L10: you can remove "quite" here (or you can keep it)

Removed.

L14: What's being driven? The clouds' existence or some specific property thereof?

Cold advection and updraft variations drive the qualitative effects of aerosols on cloud properties including Nd, CER, CLWP, and cloud lifetime. The description has been updated to: "under winter monsoon background, the qualitative influence of aerosols on cloud properties is modulated by variations in updrafts and cold advection (characterized by LTS) driven by atmospheric circulation that co-varies with aerosol concentrations ".

F2: I would personally plot na and CLWP on log scale and I would ensure the same axis is used (so that the reader can see how much lower Na and CLWP will be in the clean case)

We have redrawn Figure 2, with each subplot presenting a comparison between two experiments for a single variable.

T1: You might as well also list the microphysics you're using (SBM)...

We have added.

L189: I would say this more precisely — you're trying to avoid feedbacks into the states, right? See my first general comment

Yes, we have revised it to the more precise expression "To avoid interference from aerosol and cloud radiative feedback on ACI signals, aerosol and cloud radiative effects are disabled in the simulations" Additionally, as noted in our response to your first general comment, we have provided a more detailed

justification for this in Section 3.3 regarding the robustness of the simulated signals.

L196: Can you say more about this assimilation method?

We have added the description of the assimilation method in the final paragraph of Section 2.2.

L208: I don't really see the "positive impact" — in general, what do you mean by "positive" here? Like improvement compared to observations? Second, I don't see any significant movement in F3. The results are pretty good anyway, so it seems the assimilation made little difference and it wasn't needed... But either way, I think discussing the assimilation is confusing because the results look pretty decent without it (so there's no reason for it)

Sorry for misleading, here we wish to express that assimilation generally improves simulated meteorological fields. We have revised the descriptions in the revised manuscript.

Figure 3 in the revised manuscript shows that assimilation enhances simulations of some meteorological elements but also widens the gap between simulations and observations in many vertical layers. Based on this, the statement of the overall "positive" effect lacks conviction. We revised this paragraph to clarify that, from the perspective of vertical profiles, assimilation both increases and decreases the observation-simulation differences in meteorological elements, and we used the clear improvement in the observation-simulation correlation shown in Figure S3 to demonstrate its overall enhancing effect.

This statement is required for evaluating meteorological field simulations and elucidating assimilation effects. Additionally, the role of assimilation in ensuring consistency of meteorological conditions across different experiments and improving the robustness of simulated ACI signals is also briefly noted here.

L217: Define "Control\_NoDA" somewhere before you use it; I assume you mean control without data assimilation, right?

Yes. We have added the definition of "Control\_NoDA" in the first paragraph of Section 3.2 and in the figure caption of Figure 3.

L252: That's good, I like the fact that you did this. Question though: did you also do the experiment with the aerosol/cloud radiation effects enabled? Did you see anything interesting? I guess I am asking if you could tell us precisely what you got out of disabling these effects ...

Thank you for this comment. We conducted sensitivity tests with cloud (Rad\_CLD), aerosol (Rad\_AER), and both (Rad\_ALL) radiation effects enabled to quantify their impacts on ACI, as

detailed in our response to your first general comment and illustrated in Figure R1.

Overall, radiation effects exert minor influence on the trends of ECO meteorological-aerosol-cloud variations, but they do impact short-term changes during some periods. During these periods, the regional average changes in  $N_a$ ,  $N_d$ , and CLWP induced by radiation effects can exceed 20%. On the ACI signal, in quantitative terms, the cloud radiation effect accelerates the increase of  $N_d$  with  $N_a$  and the increase of CLWP with RH, whereas the aerosol radiation effect acts in opposition. Qualitatively, ARI increases the uncertainty in the relationship between LTS and ACI.

By disabling aerosol and cloud radiation effects in our simulations, we avoid the quantitative shifts and heightened qualitative uncertainty in ACI signals caused by ARI, thereby enabling a more accurate analysis of how meteorological conditions influence ACI across different environments.

L264: replace "is the containing of continental aerosols" with "is using continental aerosols" Replaced.

F5: good job on this figure :) just remind the reader the sampling frequency/averaging of the data in the caption

Thank you. We have reviewed and revised all figure captions to ensure that the data processing methods for each figure are appropriately described.

F9: I am not sure what this figure is showing exactly. Can you explain it more in the text? What are you trying to show us?

Figure 9 presents the synergistic variation of cloud properties under two pollution regimes in response to meteorological and aerosol conditions.

Previous analyses of cloud property variations with meteorological conditions, based on nearly 2000-fold differences in Na between the two pollution regimes, demonstrated how ACI varies with dynamic and atmospheric humidity conditions. In Section 4.2, we further analyze the role of meteorological conditions in ACI under different environments, primarily based on Figure 9. Specifically, we focus on three issues: (1) what role do meteorological conditions play in ACI variations with aerosols? (2) How much impact can changes in meteorological conditions exert on ACI under given aerosol conditions? (3) How does the sensitivity of ACI to meteorological and aerosol conditions change with environmental variations?

We have revised both the text and paragraph structure of Section 4.2 to provide a clearer discussion centered around Figure 9.

F10: As Reviewer 1 indicated, be careful with how you define the units.

Thanks for the reminder. We have corrected the data in Figure 10 and changed the figure to size distribution plots with dN/dlogD (cm-3) as the vertical axis.

F11: Consider reworking this figure so that the majority of it is not white space (same for F9)

We have redrawn Figures 9 and 11, adjusting the axes to minimize excessive blank space.

**References**

- Bennartz, R., Fan, J., Rausch, J., Leung, L. R., and Heidinger, A. K.: Pollution from China increases cloud droplet number, suppresses rain over the East China Sea, Geophys. Res. Lett., 38, <a href="https://doi.org/10.1029/2011GL047235">https://doi.org/10.1029/2011GL047235</a>, 2011.
- Fan, J. W., Leung, L. R., Li, Z. Q., Morrison, H., Chen, H. B., Zhou, Y. Q., Qian, Y., and Wang, Y.: Aerosol impacts on clouds and precipitation in eastern China: Results from bin and bulk microphysics, J. Geophys. Res.: Atmos., 117, https://doi.org/10.1029/2011jd016537, 2012.
- Fan, J., Wang, Y., Rosenfeld, D., and Liu, X.: Review of Aerosol–Cloud Interactions: Mechanisms, Significance, and Challenges, J. Atmos. Sci., 73, 4221-4252, <a href="https://doi.org/https://doi.org/10.1175/JAS-D-16-0037.1">https://doi.org/https://doi.org/10.1175/JAS-D-16-0037.1</a>, 2016.
- Guo, J., Luo, Y., Yang, J., Furtado, K., and Lei, H.: Effects of anthropogenic and sea salt aerosols on a heavy rainfall event during the early-summer rainy season over coastal Southern China, Atmos. Res., 265, 105923, <a href="https://doi.org/10.1016/j.atmosres.2021.105923">https://doi.org/10.1016/j.atmosres.2021.105923</a>, 2022.
- Haghighatnasab, M., Kretzschmar, J., Block, K., and Quaas, J.: Impact of Holuhraun volcano aerosols on clouds in cloud-system-resolving simulations, Atmos. Chem. Phys., 22, 8457-8472, https://doi.org/10.5194/acp-22-8457-2022, 2022.
- Hudson, J. G. and Noble, S.: CCN and Vertical Velocity Influences on Droplet Concentrations and Supersaturations in Clean and Polluted Stratus Clouds, J. Atmos. Sci., 71, 312-331, <a href="https://doi.org/10.1175/JAS-D-13-086.1">https://doi.org/10.1175/JAS-D-13-086.1</a>, 2014.
- Jia, H., Ma, X., Yu, F., Liu, Y., and Yin, Y.: Distinct impacts of increased aerosols on cloud droplet number concentration of stratus/stratocumulus and cumulus, Geophys. Res. Lett., 46, 13517-13525, <a href="https://doi.org/10.1029/2019GL085081">https://doi.org/10.1029/2019GL085081</a>, 2019.
- Jia, H., Quaas, J., Gryspeerdt, E., Böhm, C., and Sourdeval, O.: Addressing the difficulties in quantifying droplet number response to aerosol from satellite observations, Atmos. Chem. Phys., 22, 7353-7372, <a href="https://doi.org/10.5194/acp-22-7353-2022">https://doi.org/10.5194/acp-22-7353-2022</a>, 2022.
- Khain, A. P., Beheng, K. D., Heymsfield, A., Korolev, A., Krichak, S. O., Levin, Z., Pinsky, M., Phillips, V., Prabhakaran, T., Teller, A., van den Heever, S. C., and Yano, J. I.: Representation of microphysical processes in cloud-resolving models: Spectral (bin) microphysics versus bulk parameterization, Rev. Geophys., 53, 247-322, <a href="https://doi.org/10.1002/2014rg000468">https://doi.org/10.1002/2014rg000468</a>, 2015.
- Liu, Z., Ming, Y., Zhao, C., Lau, N. C., Guo, J., Bollasina, M., and Yim, S. H. L.: Contribution of local and remote anthro pogenic aerosols to a record-breaking torrential rainfall event in Guangdong Province, China, Atmos. Chem. Phys., 2 0, 223-241, https://doi.org/10.5194/acp-20-223-2020, 2020.
- Liu, Y., Lin, T., Zhang, J., Wang, F., Huang, Y., Wu, X., Ye, H., Zhang, G., Cao, X., and de Leeuw, G.: Opposite effects of aerosols and meteorological parameters on warm clouds in two contrasting regions over eastern China, Atmos. Chem. Phys., 24, 4651-4673, https://doi.org/10.5194/acp-24-4651-2024, 2024.
- Ma, X., Jia, H., Yu, F., and Quaas, J.: Opposite aerosol index cloud droplet effective radius correlations over major industrial regions and their adjacent oceans, Geophys. Res. Lett., 45, 5771-5778,

- https://doi.org/10.1029/2018GL077562, 2018.
- Reutter, P., Su, H., Trentmann, J., Simmel, M., Rose, D., Gunthe, S. S., Wernli, H., Andreae, M. O., and Pöschl, U.: Aerosoland updraft-limited regimes of cloud droplet formation: influence of particle number, size and hygroscopicity on the activation of cloud condensation nuclei (CCN), Atmos. Chem. Phys., 9, 7067-7080, <a href="https://doi.org/10.5194/acp-9-7067-2009">https://doi.org/10.5194/acp-9-7067-2009</a>, 2009.
- Saleeby, S. M., Berg, W., van den Heever, S., and L' Ecuyer, T.: Impact of Cloud-Nucleating Aerosols in Cloud-Resolving Model Simulations of Warm-Rain Precipitation in the East China Sea, J. Atmos. Sci., 67, 3916-3930, <a href="https://doi.org/10.1175/2010JAS3528.1">https://doi.org/10.1175/2010JAS3528.1</a>, 2010.
- Salma, I., Thén, W., Vörösmarty, M., and Gyöngyösi, A. Z.: Cloud activation properties of aerosol particles in a continental Central European urban environment, Atmos. Chem. Phys., 21, 11289-11302, <a href="https://doi.org/10.5194/acp-21-11289-2021">https://doi.org/10.5194/acp-21-11289-2021</a>, 2021.
- Wang, Z., Huang, X., Wang, N., Xu, J., and Ding, A.: Aerosol-Radiation Interactions of Dust Storm Deteriorate Particle and Ozone Pollution in East China, J. Geophys. Res.: Atmos., 125, e2020JD033601, https://doi.org/10.1029/2020JD033601, 2020.
- Zhao, Y., Gao, Y., Xu, L., and Zhang, M.: Numerical Analysis of Aerosol-Radiation Interactions and Aerosol-Cloud Interactions Impacts on Surface Ozone in Eastern China During a Pm2. 5-O3 Co-Pollution Episode, Available at SSRN 5269238, http://dx.doi.org/10.2139/ssrn.5269238, 2025.
- Zheng, X., Xi, B., Dong, X., Wu, P., Logan, T., and Wang, Y.: Environmental effects on aerosol–cloud interaction in non-precipitating marine boundary layer (MBL) clouds over the eastern North Atlantic, Atmos. Chem. Phys., 22, 335-354, <a href="https://doi.org/10.5194/acp-22-335-2022">https://doi.org/10.5194/acp-22-335-2022</a>, 2022.

---

## Author Comment (AC2)

**Response to the Comments of Referees**

**Journal:** Atmospheric Chemistry and Physics **Manuscript Number:** egusphere-2025-2555

Title: How meteorological conditions influence aerosol-cloud interactions under different pollution

regimes

Author(s): Jianqi Zhao, Xiaoyan Ma, Johannes Quaas, and Tong Yang

We thank the reviewers and the editor for providing helpful comments to improve the manuscript. We have revised the manuscript according to the comments and suggestions of the referees.

The referee's comments are reproduced (black) along with our replies (blue). All the authors have read the revised manuscript and agreed with the submission in its revised form.

**Anonymous Referee #2**

**Review of 'How meteorological conditions influence aerosol-cloud interactions under different pollution regimes'**

This study uses WRF-Chem to simulate marine liquid clouds near the Eastern China Ocean during a selected winter period. The experiments are designed to examine aerosol effects on cloud and precipitation by comparing clean versus polluted scenarios. Results show that aerosols extend cloud lifetime in moist conditions but shorten it in dry conditions. In the polluted regime, continental aerosols lead to higher Nd and CLWP and smaller CER, while precipitation responses are mixed overall. Under the clean regime, aerosol activation is efficient and responses are clearer with supersaturation. In polluted conditions, clouds exhibit mixed responses tied to different cloud processes, reflecting regime-dependent ACI sensitivity.

While the overall narrative and logic read fine, I found multiple occasions where the interpretation and discussion are rather vague or misleading. As such, I have questions and comments on technical and interpretive details. The comments below include several general comments and more specific comments, laid out in the narrative sequence when reading the manuscript.

We deeply appreciate the reviewer's constructive comments and useful suggestions. We have carefully considered the concerns regarding interpretation and discussion, and have refined the manuscript accordingly. Our detailed point-by-point responses are provided below (in blue).

**General Comments**

In the Introduction, please situate this work within prior LES efforts (WRF-SBM, WRF-Chem, SAM, etc.) over the East China Ocean and synthesize what they concluded about ACI. Then clarify the remaining gap and why a chemistry-aware framework is needed here: specifically, what does WRF-Chem capture that a physics-only perturbation of aerosol number in WRF/WRF-SBM cannot? Because much of your analysis emphasizes dynamical/thermodynamic linkages, please consider discussing

how chemical pathways modify those linkages and how your design advances beyond prior 'simple number-perturbation' studies.

We conducted a review on the ACI modeling studies over the East China Ocean and found that although ACI is a research hotspot, there are not many studies with the research area set in the East China Ocean, and most are based on a chemistry-aware framework. For example, Saleeby et al. (2010) used a cloud-resolving model to explore the response of cloud and precipitation microphysics to aerosols, while Bennartz et al. (2011) and Guo et al. (2022) used WRF-Chem to investigate the impact of continental pollution on ECO cloud microphysics and the impacts of anthropogenic and sea salt aerosols on a heavy rainfall event, respectively. These studies pointed out the increase of Nd and the decrease of precipitation caused by continental aerosols, as well as the reduction of Nd and the increase of CER due to large particles suppressing the activation of small particles. In our study, the analysis of the effects of continental aerosols generally agrees with the former, but by tracing ACI under different environments, we pointed out cases in which continental aerosols lead to enhanced precipitation. As for the latter, we did not separately analyze the impacts of large and small particles, but the revealed suppression of small particle activation by a large number of medium-sized droplets under high supersaturation, resulting in a decrease of Nd and an increase of CER, is similar to their findings.

We broadened our review objectives and, based on relevant ACI modeling studies, explained why a chemistry-aware framework is needed. Through physics-only perturbation of aerosol, related studies, on the basis of the well-established fundamental theories of ACI, explored the dependence of aerosol activation on aerosol properties and meteorological conditions (Reutter et al., 2009; Tang et al., 2024), the variation of the Twomey effect with cloud dynamics and pollution levels (Andrejczuk et al., 2014; Prabhakaran et al., 2023; Tang et al., 2024), as well as the variation of rapid adjustments with cloud properties (Heikenfeld et al., 2019; Jiang et al., 2023; Prabhakaran et al., 2023; Tang et al., 2024). This method effectively isolates the influence of external factors (such as large-scale dynamics, radiation, chemical processes, and emissions), diagnoses the response of cloud microphysics to aerosols, and deepens the understanding of ACI. However, studies based solely on physics-only perturbation of aerosol have certain limitations, for example: (1) difficulty in reproducing the spatiotemporal variation of aerosol properties (such as concentration, size distribution, and chemical composition), which may lead to distortion of activation simulations (George et al., 2015; Hodzic et al., 2023); (2) the absence or inaccuracy in treating aerosol-related physical processes (such as resuspension, wet deposition, and thermodynamic and dynamic changes induced by aerosols) may result in quantitative or even qualitative biases in ACI signals (Ahmadov, 2016; Briant et al., 2017); (3) difficulty in direct comparison with observations, leading to insufficient verifiability of the results.

Compared with physics-only perturbation of aerosol, the chemistry-aware framework can effectively address the issues of accuracy in aerosol treatment and verifiability. Based on the chemistry-aware framework, related studies have conducted extensive quantitative and qualitative research on ACI (Saleeby et al., 2010; Bennartz et al., 2011; Liu et al., 2020; Guo et al., 2022; Haghighatnasab et al., 2022), providing strong support for our understanding of the impacts of aerosols on clouds and the influencing factors of ACI under factual scenarios. However, these studies also have some limitations:

(1) compared to physics-only perturbation of aerosol, their ACI analyses introduce external interferences such as radiation and aerosol-induced meteorological changes; (2) analyses of ACI sensitivity to individual factors are often confounded by co-varying influences, leading to uncertainties in signal separation; (3) quantitative and qualitative conclusions derived from case studies may not hold under different conditions; and (4) bulk microphysical parameterizations may distort ACI signals, both quantitatively and qualitatively. In this study, we controlled the uncertainties arising from the above issues from the perspectives of technical approaches, experimental design, and analytical methods: (1) avoidance of ARI interference in ACI analyses through disabling radiative effects in simulations and ensuring consistency of meteorological conditions across experiments through fourdimensional data assimilation; (2) examination of ACI responses to the co-variation of different meteorological factors and of meteorological factors and aerosols and enhancement of ACI signal detectability through large Na differences between Control and Clean experiments; (3) selection of a case exhibiting significant co-variations in meteorology, aerosols, and clouds to support a relatively comprehensive analysis; and (4) use of high-resolution spectral bin microphysics modeling to better represent realistic ACI processes. These efforts help to deepen the understanding of the impacts of meteorological conditions on ACI and further constrain the uncertainties of ACI.

We added the reasons for selecting WRF-Chem-SBM in the third and fourth paragraphs of the Introduction, and provide a detailed description of our design in Section 3.3.

Subsection 3.2, 'Experiment setup', should be moved into the WRF-Chem description or at least earlier in Section 3, preferably before Figure 2.

It has been moved to Section 2 as Subsection 2.2.

If I am understanding correctly, the middle columns in both Figs. 4 and 5 are from the Control experiment. If so, please clarify why the spatial distributions for Nd and CLWP in Fig. 5 differ noticeably from those in Fig. 4 (e.g., an empty area in Fig. 4 but not in Fig. 5, etc.), and why precipitation also exhibits subtle differences. Please ensure consistent sampling/calculation for region maps like these.

Yes, the middle columns in Figures 4 and 5 are both from the Control experiment.

The difference arises because Fig. 4 employs simulated data that is spatio-temporally matched with observational data. The specific procedures are as follows: 1) as stated in the first paragraph of Section 2.3, when evaluating simulation data, interpolation of simulation/observation data was performed following the principle of resolution from high to low. MODIS cloud data (1 km resolution) have a higher spatial resolution than the simulation (2.4 km), while IMERG precipitation data (0.1° resolution) have a lower resolution than the simulation. Therefore, MODIS cloud data were interpolated to the model grid, while simulated precipitation was interpolated to the IMERG grid. 2) As described in the second paragraph of Section 2.3, observational and simulated data were matched for the evaluation.

Specifically, a simulated value at a given grid point and time step was considered valid only if a corresponding valid observational value was available at the same location and time. If no valid observations were available at a grid point throughout the entire study period, that location was left blank in the figure.

The simulated values used in Fig. 4 underwent the above processing steps. This rigorous matching and masking procedure explains the noticeable differences between the data presented in Fig. 4 and the study-period averages shown in Fig. 5. Additionally, since IMERG has a resolution of 1 day, Figure 4 shows the cumulative precipitation from February 2 to 4, while Figure 5 displays the cumulative precipitation for the entire simulation period (from 12:00 on February 1 to 00:00 on February 5). We have noted this in the caption of Figure 4.

When describing figure results, please also list a plain-number expression alongside the scientific expression to help intuitive interpretation. For instance, at L402: 'At sub-moderate supersaturation levels (...)' include the normal expressed percentage values ( $<10-0.9\% \rightarrow < \sim0.12\%$ ).

We have added plain-number expressions alongside all scientific expressions.

Please carefully revisit whether the cited supporting figures actually show and support the associated statements. Some cases are included in detail comments below.

We have made revisions based on your specific comments and carefully reviewed and modified the entire manuscript to ensure that cited supporting figures support the associated statements.

**Specific Comments**

L47. You may want to narrow the statement down to 'satellite observation...' given the reference cited.

Here we aim to express the general limitations of observations, including retrieval uncertainties, instrument errors, and interference from the radiative effects of aerosols and clouds, thereby introducing the need for the integration of simulation models. The original citation only supported uncertainty in satellite retrieval and interference from radiation, which was insufficient. We added a cited statement regarding instrument errors.

**L62.** Please include a brief description of the following sessions of the manuscript.

We added a brief description of the sections at the end of the first section.

**L96.** What is the uncertainties of this Nd retrieval compared with the in-situ observation.

The studies evaluated the  $N_d$  retrievals against in-situ observation, and found that the  $N_d$  retrievals perform well for relatively homogeneous, optically thick and unobscured stratiform clouds under high solar zenith angle conditions (Grosvenor and Wood, 2014; Bennartz and Rausch, 2017; Jia et al., 2021). At very thin cloud, the retrievals suffer from large uncertainties arising from instrument errors and other sources of reflectance error for COT and CER (Zhang and Platnick, 2011; Sourdeval et al., 2016), we thus exclude the data for transparent-cloudy pixels (i.e. COT < 5), as described in Section 2.4.

We added this description in Section 2.3.

L99. Please specify the (approximate) local hour of Terra overpasses over ECO.

It passes over ECO at approximately 03:00 UTC, which corresponds to around 11:00 local time. We have added notes here.

**L111.** Please specify the matching logic (e.g., whether the WRF output at the nearest time step or an average over surrounding times was used). For a given WRF grid cell, approximately how many MODIS L2 records were averaged?

The matching utilizes the nearest time step of WRF output. During the study period, the number of valid MODIS records varies across grid cells, with the averaged value for each cell corresponding to 0 to 6 records. This description has been added to the fourth paragraph of Section 2.4.

**L126.** How much uncertainty will these two different modeled Nd derivations introduce into the later ACI analysis? Can you show a comparison of calculated/simulated Nd vs. directly output Nd?

Based on the results of the Control experiment, we present the  $N_d$  values derived from both the empirical formula calculation and the direct model output (Fig. R2).

Physically, the model-output  $N_d$  is the result of precise calculations from the model's prognostic equations and is highly consistent with meteorological, aerosol, and other cloud properties. In contrast, the  $N_d$  calculated using the empirical formula is employed to enable comparison with satellite data on the same basis, but its physical consistency and accuracy are inferior to the model-output values. In terms of overall magnitude, because a relatively loose criterion (CIWC = 0 and CLWC > 0) was adopted for identifying liquid-phase cloud layers in the simulations, many tenuous clouds that would not be detected by satellites were included in the in-cloud mean calculation. Consequently, the model-output  $N_d$  is, on average, one order of magnitude lower than that calculated using the empirical formula.

Regarding uncertainty, after dividing the empirically calculated  $N_d$  by 10 to offset the systematic numerical difference between the two  $N_d$ , the two show similar variations at high COT (Figs. R2a–c). However, for optically thin clouds (COT < 5.0), the empirical formula tends to overestimate  $N_d$ , a finding that aligns with the evaluation of satellite  $N_d$  retrievals cited in Section 2.3 and is consistent

with the rationale for the MODIS data screening criteria. The use of empirically derived  $N_d$  introduces substantial uncertainty into the associated ACI analysis. Specifically, for clouds with  $N_d < 1000$  cm-3, this can amplify or reduce ACI signals directly related to  $N_d$  by factors ranging from 2 up to more than 10, with the impact increasing as  $N_d$  decreases (Fig. R2f, i, and l). Given the large uncertainties introduced by the empirical formula in ACI analysis, the model-output  $N_d$  is used for the analyses based on simulation results. We have added this note to the third paragraph of Section 2.4.

**Figure R2.** In-cloud mean Nd (Control experiment) derived from direct model output (left column) and empirical formula (Eq. (1) in the main text, middle column), along with the ratio of (b) to (a) divided by 10 (right column), as functions of COT and CER (a–c), LTS and RH (d–f), Na and LTS (g–i), and Na and RH (j–l). The data processing method is the same as that used in Fig. 6.

**L141.** Please specify the lat/lon range of the red box.

The ECO region in this study spans longitudes 123.03–126.24°E and latitudes 30.41–33.14°N. We added this note where we first defined the simulated ECO domain in Section 2.2.

L152. Is there a specific reason for selecting 1300 m? Also, how does the model cloud-top height compare with the satellite CTH?

Since this study focuses on liquid-phase clouds, which typically reside at low altitudes, using column-averaged meteorological variables cannot accurately capture the influence of meteorological fields on ACI. Therefore, we adopted the column-averaged value of the lower atmosphere. The specific choice of 1300 m corresponds to the mean height of the model's 16th layer (counting from the surface upward), as layers 1–16 encompass over 97% of cloud droplets across all experiments. The meteorological conditions within this range are most closely linked to ACI in the liquid-phase clouds studied.

In Section 3.2, we added evaluations of simulated CTH and cloud top temperature (CTT). The simulated and observed CTH show strong correlation (r = 0.83) in spatial distribution. However, due to differences in cloud-top detection methods between the model and MODIS, as well as limitations in the model's ice-phase parameterization (many MODIS liquid-cloud samples exhibit CTT below -  $30^{\circ}$ C, while the simulated liquid clouds have a minimum CTT of -7°C. Under the model's parameterization, only mixed-phase or ice-phase clouds exhibit lower CTT values), the model underestimates CTH by an average of 53.71% compared to MODIS.

L152. Please correct me if I'm wrong, but by eye are LTS and RH exactly the same between the Control and Clean regimes, or are the differences too subtle to visualize?

We have redrawn Figure 2 so that each subplot presents a comparison between two experiments for a single variable, thereby more clearly illustrating the variations of individual variables between them. As shown in the new Figures 2a and 2b, the LTS and RH from the two experiments exhibit general consistency in both magnitude and temporal evolution. This aligns with our intended outcome; i.e., by assimilating observations and turning off radiative effects, external factors are minimized, thereby ensuring that the differences in ACI between the two experiments arise primarily from the influence of continental aerosols.

At a more detailed level in the new figures, slight discrepancies are observed in the mean values and uncertainty ranges of LTS and RH between the two experiments, which are attributed to feedback from aerosol–cloud interactions on the meteorological fields.

L154. Do the variables shown in Fig. 2 represent domain means or cloud-sample means?

Fig. 2 represents domain means. We have added the note "averaged over ECO" to the figure caption.

L155. This 'dominant-factor transition' statement needs physical-mechanism support. Otherwise, omit it and reframe as a description of wind direction (as in the next sentence), or simply state that the domain was impacted by cold-air advection from the northwest.

We have revised this sentence to the relatively rigorous description "Changes in atmospheric circulation alter the meteorological conditions over ECO, as characterized by variations in LTS." to ensure coherence with the surrounding context.

L162. ...inhibiting updraft (Fig. S1i)

Thanks for the reminder. We've added it.

**L163.** I wouldn't say 'rapid' given the wind speed isn't significantly faster than in the LTS < 17 scenario.

We have changed it to "...due to intense condensation caused by the advection of northwestern low-level cold air...".

L171. 'low-pollution' might be misleading, try 'moderate' or 'relatively low', given that 'clean' is used later.

We have changed it to "relatively low-pollution".

L172. That is an ultraclean condition; I would like to see the corresponding Nd.

We have added Nd to Figure 2 and its corresponding description.

L173. Which case? Are you referring to the polluted or clean case as reflecting typical variations in the joint field over your study region? Please clarify. I recommend retuning this paragraph so the message is tied more tightly to this particular study if you want to keep these general statements.

The term "case" here denotes the ACI across both the Control and Clean experiments during the study period. It captures the clear variations in meteorological conditions, the distinct aerosol changes within each experiment, and the significant aerosol differences between the two simulations—collectively representing typical variations in aerosols and meteorological fields. Our previous statement was not sufficiently clear. We have revised this paragraph to the more precise and contextually relevant

statement "As mentioned above, the shift in meteorological conditions over ECO during the simulation period, the distinct aerosol variations in each experiment, the significant aerosol differences between the two experiments, and the resulting disparities in cloud properties—both within each experiment and between them—collectively provide abundant samples for ACI analysis. This robustly supports the comprehensiveness of the study".

**L211.** It looks like the Control has larger error bars (and slightly more outliers) than Control\_NoDA, especially for the wind components. I recognize the better correlation in Fig. S2; however, can you quantify mean bias or RMSE between these two and observations to support your statement?

The impact of assimilation on the NMB and RMSE between observations and simulations varies across meteorological elements, with both positive and negative effects observed. These two metrics do not sufficiently support the claim of an "overall positive impact".

We have revised our previous statement regarding the "overall positive impact" based on the vertical profiles in Figure 3, as it lacked rigor. The updated text acknowledges that, from the perspective of statistics at each vertical level, data assimilation improves the simulation of some meteorological elements while also exacerbating model-observation discrepancies in many cases. The overall enhancing effect of assimilation is instead supported by the clearly improved model-observation correlation demonstrated in Figure S3. Additionally, we briefly note here (with a detailed explanation provided in Section 3.3) that assimilation helps maintain meteorological consistency between experiments, to assist in illustrating the necessity of assimilation.

**L240.** Please also note in the caption that the lower subpanels are for the red-box domain.

We have added the note indicating the domains corresponding to the subpanels in the figure caption.

**L297.** If you only present counts in Fig. 6m-o, I would not call it 'occurrence frequency', which usually meant for depict fraction or percentage. Suggest just use 'counts' or 'samples'. Or maybe you can get the actual fractional frequency of occurrence of clouds.

We have replaced "frequency" with "counts" in the relevant figures and their corresponding descriptions.

**L301.** Do you have a physical explanation for the increased cloud lifetime at the high-LTS tail (>24 K), which also corresponds to the lowest RH? I wonder if entrainment mixing is inhibited, given that Nd (CER) is comparatively lower (larger) at that tail.

Thank you very much for this comment. We compared the entrainment intensity of the Control

experiment between LTS  $\geq$  24 K and LTS < 24 K conditions (Fig. R3a), and between the Control and Clean experiments under LTS  $\geq$  24 K conditions (Fig. R3b) under relatively dry conditions (RH < 80%), to explain the increased cloud lifetime at the high-LTS tail (>24 K). By adapting the methodology of Jia et al. (2019) and refining it based on model outputs, we defined the entrainment intensity as the difference in CLWC between the layers immediately below (non-entrainment zone) and above (entrainment zone) the vertical layer with maximum CLWC in clouds with CLWP > 1 g·m-2, divided by the maximum CLWC (i.e. (CLWCnon-entrainment – CLWCentrainment) / CLWCmax).

Figure R3. Frequency distribution histograms and fitted curves of entrainment intensity for liquid-cloud samples (CLWP >  $1~\rm g\cdot m^{-2}$ ) under relatively dry conditions (RH < 80%), showing comparisons between environments with LTS  $\geq 24~\rm K$  and LTS <  $24~\rm K$  (a, from Control experiment), and between Control and Clean experiments within the LTS  $\geq 24~\rm K$  environment (b). The entrainment intensity is defined, following Jia et al. (2019) with refinements based on model outputs, as (CLWCnon-entrainment – CLWCentrainment) / CLWCmax, where the non-entrainment and entrainment zones are the layers immediately below and above the level of maximum CLWC in clouds with CLWP >  $1~\rm g\cdot m^{-2}$ . The histograms are obtained by first assigning liquid-phase cloud samples under different conditions to entrainment intensity intervals according to their corresponding entrainment intensity, and then calculating, for each interval, the relative frequency of samples with respect to the total number of samples. The curves represent fitted distributions of these frequencies to better highlight the differences in entrainment intensity under different conditions.

The results align with your inference: entrainment intensity consistently decrease both in high-LTS environments compared to low-LTS conditions and in Control relative to Clean experiments under high-LTS regimes. This demonstrates that stable atmospheric conditions and enhanced cloud development driven by continental aerosols suppress the weakening of clouds by entrainment, ultimately prolonging cloud lifetime. We have added this analysis at the end of the second paragraph in Section 4.1.

**L335.** It will not be intuitive to say 'more large droplets,' since the 75th percentile of CER under Control is still largely smaller than under Clean. Consider: 'collision—coalescence at lower levels can produce large droplets.'

We have changed it to "The enhanced vertical development of low-level clouds (below 1500 m), as shown in Figs. 8e-h and S9g-l, strengthens collision-coalescence (producing large droplets)..."

**L336.** Have you checked cloud fraction and sample size vertically? In the Increase\_ scenario, the ~3 km samples appear fewer (narrower spread), so precipitation enhancement may stem from different cloud layers (and not only from 'vertical development of the cloud layer'). If so, please make this clear.

It would help to examine vertical cross-sections of clouds (Nd, CER, Nr, etc.) for both scenarios to confirm. Since Fig. 4 suggests regional clustering for Increase\_/Decrease\_ samples, a SW-NE slice could work, please consider this.

Since the cumulative precipitation shown in Fig. 5 exhibits an east-west distribution pattern, its zonal variation provides a relatively clear basis for stratified precipitation analysis. We therefore generated vertical cross-sections (Fig. S9) with dimensions of height and longitude, where values at each longitude coordinate represent the mean across all latitudinal grids at that longitude.

From the cross-sections of CLWC and RWC, it is clear that precipitation originates from different cloud layers. We revised this paragraph to "the enhanced vertical development of low-level clouds (below 1500 m) strengthens collision-coalescence (producing large droplets), and, coupled with a substantial increase in droplet numbers in the  $N_d$ -limited relatively high-level clouds (above 1500 m), collectively drives the pronounced rise in  $N_r$  and RWP across all levels with liquid-phase cloud".

**Figure 8.** The legends (e.g., Increase/Decrease\_CER) are misleading. Consider renaming to clearer identifiers, e.g., CER (increased-precip samples).

We have changed the legends to "...increased/decreased-precip samples".

**L364.** The correlation is significant but moderate; what message is this intended to convey? Because you did not show the relative vertical position of 950 hPa cold-air advection versus the cloud layers, the physical interplay is not determinate. I guess you can remove this...

We have removed this sentence.

**L365.** Can you remind me where did you show the Na relationship to LTS? Are you referring to Fig. 9a? And please add the support figure indentifer to this two statements.

Based on the covariation between LTS and  $N_a$  in Figure 2 and the circulation changes under different LTS states in Figure 1, we demonstrate the overall consistency between LTS and  $N_a$  over ECO during the winter monsoon in the Control experiment. This statement is repeated here to maintain paragraph continuity, but the absence of identifiers and corresponding explanations makes these assertions appear abrupt. And, the discussion here is referring to Column 2 of Fig. 9.

We have revised and added the identifier for the supporting figure in these statements. This paragraph has been moved to the end of the second paragraph in Section 4.2 to enhance the structural clarity of

the description related to Figure 9.

**Figure 10.** Please clarify the units in panel (a). The in-cloud Na magnitudes in both experiments seem to far exceed the number concentrations shown previously (e.g., Figs. 2, 5). If you are showing an aerosol size distribution, the y-axis should be dN/dlogDp (cm-3).

Also, what is the relationship between activated aerosol Naa and cloud-droplet Nd in the model outputs? Please elaborate.

Thank you for the reminder. We have corrected the data in Figure 10 and changed the figure to size distribution plots with dN/dlogD (cm-3) as the y-axis.

In WRF-Chem-SBM, each cloud droplet corresponds to an activated aerosol particle, meaning  $N_d$  and  $N_{aa}$  are equivalent in total quantity. Figure 10b shows the spectral distribution of cloud droplets through the variation of  $N_d$  across different bins, while Figure 10a displays the activation efficiency of aerosols by comparing  $N_a$  and  $N_{aa}$  of the four aerosol bins. We have added this explanation to the caption of Figure 10 and included an identifier in the corresponding text.

**L369.** Fig. 10b provides limited support for this statement (no CLWP/RLWP shown). Please elaborate and/or provide a clear reference.

We have added the reference "(Figs. 9e-f, i-j and q-r)" here.

**L377.** At least CER and RWP do not increase steadily with RH. Please provide statistical support for 'stronger sensitivity,' e.g., correlations or regression slopes of cloud properties vs. LTS/RH.

We have revised "steadily increasing" to "generally increasing" (as correlations of CER and RWP with RH being 0.46 and 0.67, respectively).

We have added a statistical analysis of correlations between cloud properties and LTS/RH (Table R1) to support the statement of "stronger sensitivity", and revised the corresponding content in the manuscript.

**Table R1.** Correlations between cloud properties (shown in Fig. 9) and LTS or RH (correlations are calculated after averaging cloud property values across all Na coordinates for each corresponding LTS or RH value).

|          | RH    |        |          |        | LTS   |        |          |        |
|----------|-------|--------|----------|--------|-------|--------|----------|--------|
| Property | Clean |        | Polluted |        | Clean |        | Polluted |        |
|          | r     | p      | r        | p      | r     | p      | r        | p      |
| CER      | 0.46  | < 0.01 | 0.96     | < 0.01 | 0.86  | < 0.01 | 0.15     | 0.13   |
| CLWP     | 0.81  | < 0.01 | 0.75     | < 0.01 | -0.54 | < 0.01 | -0.34    | < 0.01 |

| RWP                  | 0.67 | < 0.01 | -0.14 | 0.16   | 0.18  | 0.07   | -0.50 | < 0.01 |
|----------------------|------|--------|-------|--------|-------|--------|-------|--------|
| $log_{10}(N_d/N_a)$  | 0.59 | < 0.01 | 0.89  | < 0.01 | -0.33 | < 0.01 | 0.28  | < 0.01 |
| $log_{10}(CLWP/N_a)$ | 0.58 | < 0.01 | 0.70  | < 0.01 | -0.14 | 0.18   | -0.82 | < 0.01 |
| $log_{10}(RWP/N_a)$  | 0.65 | < 0.01 | 0.70  | < 0.01 | 0.22  | 0.03   | -0.36 | < 0.01 |

**L379.** Please elaborate. If activation ratio, CLWP, and RWP largely increase with Na across RH ranges, how is 'greater sensitivity' exhibited?

We have used the correlations presented in Table S1 to demonstrate that under the polluted regime, CER, Nd/Na, CLWP/Na, and RWP/Na all exhibit higher sensitivity to RH (on average over 46.98% higher) compared to the clean regime. This contrasts with the behavior of CLWP and RWP, which show slightly and significantly lower sensitivity to RH, respectively, than under the clean regime, due to interference from their dependence on Na caused by these collision-coalescence-dominated cloud properties.

We have revised the statement in this section (the third paragraph of Section 4.2) based on the statistics.

**L380.** Regarding the 'manner' of cloud vs. Na being similar to high LTS (I presume Fig. 9b, 9f, etc.): would that mean cloud properties are largely impacted by aerosol loading rather than environmental parameters? If so, what 'dominated role' is cold-air advection playing? You have not established a strong physical relationship between cold advection and cloud (a correlation between column-mean Na and 950 hPa temperature is not enough). If you want to discuss the physical role of cold advection, at least show the vertical dependence of cloud properties on it. Please elaborate and provide figure support.

Regarding the comment on "mean cloud properties are largely impacted by aerosol loading rather than environmental parameters", we acknowledge that the results in Figure 9 do show a significant influence of  $N_a$  on cloud properties, with the rate of change in cloud properties with respect to  $N_a$  often exceeding that with respect to meteorological factors in many conditions. Our original intention was to use the broadly consistent variation of cloud properties with  $N_a$  across both RH and high-LTS perspectives to illustrate the dominant role of cold-air advection associated with high LTS in ACI under winter monsoon background. However, we recognize that this variation is substantially influenced by  $N_a$ , making our previous assertion of a "dominated role" insufficiently rigorous. Accordingly, the statement has been revised to summarize the regulatory role of meteorological fields on ACI, as demonstrated in Figure 9. Specifically, the shift from convection-dominated regimes at low LTS to cold advection-dominated regimes at high LTS modulates the qualitative impact of aerosols on cloud properties under polluted conditions, while the level of RH governs the quantitative effect of aerosols on cloud development.

Regarding the physical relationship between cold advection and clouds, Section 3.1 demonstrates the synchrony among high LTS, strong cold advection, and high Na in the Control experiment through atmospheric circulation and Na variations corresponding to different LTS levels. Section 4.1 further illustrates the association of cloud development in the Control experiment and the enhancement effect of continental aerosols on clouds with cold advection through cloud property variations with LTS. This association exists only in the Control experiment. In the Clean experiment, which lacks continental aerosols, high values of Nd and CLWP instead mainly appear under low LTS.

Furthermore, our argument in Section 4.1 regarding the dominant role of cold advection was also insufficiently rigorous, and we have revised it to "In the Clean experiment, high Nd and CLWP occur more frequently in low-LTS environments dominated by updrafts, compared to high-LTS conditions. In contrast, the Control experiment shows a significantly higher frequency of high Nd and CLWP under high-LTS conditions, resulting from the combined effects of substantial continental aerosol transport associated with winter monsoon and strong condensation, as described in Section 3.1".

**L392.** The RWP sharply decrease with Na (> 10-0.4) under low supersaturation, which reflects the precip suppression effect of aerosol, please spell it out.

We have changed this sentence to "When  $N_a$  exceeds  $10^{-0.4}$  ( $\sim 0.40$ ) cm-3, further increases in  $N_a$  under low supersaturation cause CER to rapidly decrease to around 15  $\mu$ m, thus suppressing precipitation, while high RWP appears only under high supersaturation".

**L406.** ~0.16% supersaturation is not necessarily 'high' condition.

Due to the substantial depletion of supersaturation by aerosols, although this supersaturation is not high in absolute terms, it represents a relatively high value among all Control samples. We have attached this note to the text and changed "high supersaturation" to "relatively high supersaturation".

**L407.** ~0.01% supersaturation is fairly low, again, describing the supersature with a normal expression of number will give the reader more sense.

We have revised this to "relatively moderate supersaturation" and provided normal expressions alongside these scientific expressions.

**L410.** As above, please further explain the physical basis for cold-advection dominance. This statement is somewhat misleading: updraft strength has limited direct relationship to aerosol concentrations. Please specify which cloud process is dominated by updraft under low-aerosol conditions and how that dominance is dampened under high-aerosol conditions. In which experiment would cold-advection effects be more effective?

Yes, updraft strength cannot be directly attributed to aerosol concentrations. Our intended argument was based on the co-variation between LTS and  $N_a$  over ECO under winter monsoon background, as described in Section 3.1. Specifically, lower  $N_a$  generally corresponds to lower LTS, where cloud processes are dominated by updrafts, while higher  $N_a$  is associated with higher LTS, a regime in which updrafts are suppressed and cloud processes are governed by cold-air advection. However, our initial statement was somewhat ambiguous in conveying this reasoning. To clarify, we have revised the relevant paragraph to repeat the pattern of the Control experiment presented in Section 3.1: increasing LTS is accompanied by rising  $N_a$ , weakening updrafts, and strengthening cold-air advection. This interpretation is further supported by the decrease in temperature advection and vertical wind speed with increasing  $N_a$  in the Control experiment, as shown in Figure R4.

**Figure R4.** Variations of vertically weighted mean temperature advection (ADV, calculated as  $-\left(U\frac{\partial T}{\partial x} + V\frac{\partial T}{\partial y}\right)$ ) and vertical wind velocity (W) from the near-surface to 1300 m with column-averaged Na over ECO (the value in the figure represents the ECO average).

The significant impact of the synchronous intensification of cold advection and  $N_a$  on ACI is only evident in the control experiment corresponding to the polluted regime. In the clean experiments without continental aerosols, the variation in  $N_a$  shows no clear correlation with cold advection. We have revised this paragraph based on the comment.

**L411.** Provide statistical evidence or figure support for "At moderate Na, both updrafts and cold advection are weak..."

We have added statistical support for it (Fig. S10) and revised the sentence to "moderate  $N_a$  environments experience neither sufficiently strong cold advection nor sustained updrafts (as the average variations of temperature advection and vertical wind speed with  $N_a$  in ECO, shown in Fig. S10)".

**L449.** You may wish to point out that the extensions of cloud lifetime (actually cloud frequency in this study; Fig. 6) at low and high LTS arise from different physical reasons: low LTS/vigorous updrafts

invigorate frequent cloud hydrometeors, whereas high LTS/stronger subsidence maintains the cloud layer.

Many thanks for this comment. We have revised the conclusion and the corresponding statement in the second paragraph of Section 4.1. For example, the sentence in the conclusion has been changed to "Continental aerosols extend cloud lifetime (actually cloud sample counts in this study) in moist environments, with this enhancement strongest under low and high LTS conditions, corresponding to vigorous updrafts (invigorating frequent cloud hydrometeors) and strong cold advection (causing intense condensation and maintaining cloud), respectively".

**L482.** I would like to see a revised Conclusions section after you address the main-text comments.

We have revised the conclusion.

**References**

- Ahmadov, R.: WRF-Chem: Online vs Offline Atmospheric Chemistry Modeling, ASP Colloquium; National Center for Atmospheric Research (NCAR): Boulder, CO, USA, <a href="https://edec.ucar.edu/sites/default/files/2022-02/1">https://edec.ucar.edu/sites/default/files/2022-02/1</a> Ahmadov 07 29 2016.pdf, 2016.
- Andrejczuk, M., Gadian, A., and Blyth, A.: Numerical simulations of stratocumulus cloud response to aerosol perturbation, Atmos. Res., 140-141, 76-84, <a href="https://doi.org/10.1016/j.atmosres.2014.01.006">https://doi.org/10.1016/j.atmosres.2014.01.006</a>, 2014.
- Bennartz, R., Fan, J., Rausch, J., Leung, L. R., and Heidinger, A. K.: Pollution from China increases cloud droplet number, suppresses rain over the East China Sea, Geophys. Res. Lett., 38, <a href="https://doi.org/10.1029/2011GL047235">https://doi.org/10.1029/2011GL047235</a>, 2011.
- Bennartz, R. and Rausch, J.: Global and regional estimates of warm cloud droplet number concentration based on 13 years of AQUA-MODIS observations, Atmos. Chem. Phys., 17, 9815-9836, <a href="https://doi.org/10.5194/acp-17-9815-2017">https://doi.org/10.5194/acp-17-9815-2017</a>, 2017.
- Briant, R., Tuccella, P., Deroubaix, A., Khvorostyanov, D., Menut, L., Mailler, S., and Turquety, S.: Aerosol–radiation interaction modelling using online coupling between the WRF 3.7.1 meteorological model and the CHIMERE 2016 chemistry-transport model, through the OASIS3-MCT coupler, Geosci. Model Dev., 10, 927-944, <a href="https://doi.org/10.5194/gmd-10-927-2017">https://doi.org/10.5194/gmd-10-927-2017</a>, 2017.
- George, C., Ammann, M., D'Anna, B., Donaldson, D. J., and Nizkorodov, S. A.: Heterogeneous photochemistry in the atmosphere, Chem Rev, 115, 4218-4258, https://doi.org/10.1021/cr500648z, 2015.
- Grosvenor, D. P. and Wood, R.: The effect of solar zenith angle on MODIS cloud optical and microphysical retrievals within marine liquid water clouds, Atmos. Chem. Phys., 14, 7291-7321, <a href="https://doi.org/10.5194/acp-14-7291-2014">https://doi.org/10.5194/acp-14-7291-2014</a>, 2014.
- Guo, J., Luo, Y., Yang, J., Furtado, K., and Lei, H.: Effects of anthropogenic and sea salt aerosols on a heavy rainfall event during the early-summer rainy season over coastal Southern China, Atmos. Res., 265, 105923, https://doi.org/10.1016/j.atmosres.2021.105923, 2022.
- Haghighatnasab, M., Kretzschmar, J., Block, K., and Quaas, J.: Impact of Holuhraun volcano aerosols on clouds in cloud-system-resolving simulations, Atmos. Chem. Phys., 22, 8457-8472, https://doi.org/10.5194/acp-22-8457-2022, 2022.
- Heikenfeld, M., White, B., Labbouz, L., and Stier, P.: Aerosol effects on deep convection: the propagation of aerosol perturbations through convective cloud microphysics, Atmos. Chem. Phys., 19, 2601-2627, https://doi.org/10.5194/acp-19-2601-2019, 2019.

- Hodzic, A., Mahowald, N., Dawson, M., Johnson, J., Bernardet, L., Bosler, P. A., Fast, J. D., Fierce, L., Liu, X., Ma, P.-L., Murphy, B., Riemer, N., and Schulz, M.: Generalized Aerosol/Chemistry Interface (GIANT): A Community Effort to Advance Collaborative Science across Weather and Climate Models, Bull. Am. Meteorol. Soc., 104, E2065-E2080, <a href="https://doi.org/10.1175/BAMS-D-23-0013.1">https://doi.org/10.1175/BAMS-D-23-0013.1</a>, 2023.
- Jia, H. L., Ma, X. Y., and Liu, Y. G.: Exploring aerosol-cloud interaction using VOCALS-REx aircraft measurements, Atmos. Chem. Phys., 19, 7955-7971, <a href="https://doi.org/10.5194/acp-19-7955-2019">https://doi.org/10.5194/acp-19-7955-2019</a>, 2019.
- Jia, H. L., Ma, X. Y., Yu, F. Q., and Quaas, J.: Significant underestimation of radiative forcing by aerosol-cloud interactions derived from satellite-based methods, Nat. Commun., 12, <a href="https://doi.org/10.1038/s41467-021-23888-1">https://doi.org/10.1038/s41467-021-23888-1</a>, 2021.
- Jiang, M., Li, Y., Hu, W., Yang, Y., Brasseur, G., and Zhao, X.: Model-based insights into aerosol perturbation on pristine continental convective precipitation, Atmos. Chem. Phys., 23, 4545-4557, <a href="https://doi.org/10.5194/acp-23-4545-2023">https://doi.org/10.5194/acp-23-4545-2023</a>, 2023.
- Liu, Z., Ming, Y., Zhao, C., Lau, N. C., Guo, J., Bollasina, M., and Yim, S. H. L.: Contribution of local and remote anthropogenic aerosols to a record-breaking torrential rainfall event in Guangdong Province, China, Atmos. Chem. Phys., 20, 223-241, https://doi.org/10.5194/acp-20-223-2020, 2020.
- Khain, A. P., Beheng, K. D., Heymsfield, A., Korolev, A., Krichak, S. O., Levin, Z., Pinsky, M., Phillips, V., Prabhakaran, T., Teller, A., van den Heever, S. C., and Yano, J. I.: Representation of microphysical processes in cloud-resolving models: Spectral (bin) microphysics versus bulk parameterization, Rev. Geophys., 53, 247-322, https://doi.org/10.1002/2014rg000468, 2015.
- Liu, Y., Lin, T., Zhang, J., Wang, F., Huang, Y., Wu, X., Ye, H., Zhang, G., Cao, X., and de Leeuw, G.: Opposite effects of aerosols and meteorological parameters on warm clouds in two contrasting regions over eastern China, Atmos. Chem. Phys., 24, 4651-4673, <a href="https://doi.org/10.5194/acp-24-4651-2024">https://doi.org/10.5194/acp-24-4651-2024</a>, 2024.
- Prabhakaran, P., Hoffmann, F., and Feingold, G.: Evaluation of Pulse Aerosol Forcing on Marine Stratocumulus Clouds in the Context of Marine Cloud Brightening, J. Atmos. Sci., 80, 1585-1604, <a href="https://doi.org/10.1175/JAS-D-22-0207.1">https://doi.org/10.1175/JAS-D-22-0207.1</a>, 2023.
- Reutter, P., Su, H., Trentmann, J., Simmel, M., Rose, D., Gunthe, S. S., Wernli, H., Andreae, M. O., and Pöschl, U.: Aerosoland updraft-limited regimes of cloud droplet formation: influence of particle number, size and hygroscopicity on the activation of cloud condensation nuclei (CCN), Atmos. Chem. Phys., 9, 7067-7080, <a href="https://doi.org/10.5194/acp-9-7067-2009">https://doi.org/10.5194/acp-9-7067-2009</a>, 2009.
- Saleeby, S. M., Berg, W., van den Heever, S., and L' Ecuyer, T.: Impact of Cloud-Nucleating Aerosols in Cloud-Resolving Model Simulations of Warm-Rain Precipitation in the East China Sea, J. Atmos. Sci., 67, 3916-3930, <a href="https://doi.org/10.1175/2010JAS3528.1">https://doi.org/10.1175/2010JAS3528.1</a>, 2010.
- Sourdeval, O., C.-Labonnote, L., Baran, A. J., Mülmenstädt, J., and Brogniez, G.: A methodology for simultaneous retrieval of ice and liquid water cloud properties. Part 2: Near-global retrievals and evaluation against A-Train products, Q. J. R. Meteorolog. Soc., 142, 3063-3081, <a href="https://doi.org/https://doi.org/10.1002/qj.2889">https://doi.org/https://doi.org/https://doi.org/10.1002/qj.2889</a>, 2016.
- Tang, S., Wang, H., Li, X. Y., Chen, J., Sorooshian, A., Zeng, X., Crosbie, E., Thornhill, K. L., Ziemba, L. D., and Voigt, C.: Understanding aerosol cloud interactions using a single-column model for a cold-air outbreak case during the ACTIVATE campaign, Atmos. Chem. Phys., 24, 10073-10092, <a href="https://doi.org/10.5194/acp-24-10073-2024">https://doi.org/10.5194/acp-24-10073-2024</a>, 2024.
- Zhang, Z. and Platnick, S.: An assessment of differences between cloud effective particle radius retrievals for marine water clouds from three MODIS spectral bands, J. Geophys. Res.: Atmos., 116, <a href="https://doi.org/https://doi.org/10.1029/2011JD016216">https://doi.org/https://doi.org/10.1029/2011JD016216</a>, 2011.

---

## Author Response (AR2)

**Response to the Comments of Referees**

**Journal:** Atmospheric Chemistry and Physics
**Manuscript Number:** egusphere-2025-2555
**Title:** How meteorological conditions influence aerosol-cloud interactions under different pollution regimes
**Author(s):** Jianqi Zhao, Xiaoyan Ma, Johannes Quaas, and Tong Yang

We thank the reviewers and the editor for providing helpful comments to improve the manuscript. We have revised the manuscript according to the comments and suggestions of the referees.
The referee's comments are reproduced (black) along with our replies (blue). All the authors have read the revised manuscript and agreed with the submission in its revised form.

**Anonymous Referee #2**

I thank the authors for their substantial effort in addressing the reviewers' comments and revising the manuscript. The added and revised figures, together with the expanded discussion, satisfactorily resolve my earlier concerns and lead to statistically robust results and more accurate arguments. The manuscript now makes more precise claims and appropriately acknowledges both the strengths and the limitations of the study, providing food for thought. I have only a few minor (mostly copy-editing) comments below. I recommend publication after these are addressed.

We sincerely thank you for your meticulous review of our manuscript and for the highly valuable comments and suggestions you provided. With the help of you and the other referee, we have addressed many shortcomings in the manuscript, effectively enhanced the rigor of our arguments, and significantly improved the overall quality of the paper. Regarding the new comments you raised, we have carefully checked and revised them, as detailed in the point-by-point response below.

Minor Comments

I'm not sure the frequent use of em-dashes in the text conforms to ACP style. Please confirm this during typesetting and carefully self-review the generated content.

We consulted the ACP guidelines regarding the use of em-dashes, as follows: "En dashes (–) are longer than hyphens (-) and serve numerous purposes. We recommend spaced en dashes for syntactic constructions, not em dashes (—). En dashes are used to indicate, among other things, relationships (e.g. ocean–atmosphere exchange), ranges (e.g. 12–20 months), and components of a mixture (e.g. dissolved in 5:1 glycerin–water). They are also used to link the names of two or more persons used as a modifier (e.g. Stefan–Boltzmann constant).".

Since we are currently unsure whether the em-dashes in our manuscript have been used inappropriately, we have not made any changes to them at this stage and will adjust them during the typesetting process according to the editor's requirements.

Figure 4. As per your responses, Nd, LWP, CTH, and CTT are also sampled using rigorous thresholds, right? Here you only mention precipitation. Please adjust the caption accordingly.

We have added the following note at the end of the caption for Figure 4: "The simulated data in the figure are processed using interpolation and spatio-temporal matching (the latter is applied only for comparison with satellite-retrieved data exhibiting spatio-temporal discontinuities), as detailed in Section 2.4. For each grid cell, the corresponding simulated result is displayed only when observational data is available".

Also, consider adding this statement, as in the response, to the main text (if not already included): "Specifically, a simulated value at a given grid point and time step was considered valid only if a corresponding valid observational value was available at the same location and time."

The fourth paragraph of Section 2.4 in the main text already includes this content: "Specifically, only when satellite data at a given grid cell and time is valid, the corresponding model output (at the nearest time step to the satellite data at the same grid cell) is included in the average calculation".

Figure 5. Please indicate in the caption that the quantities are study-period averages.

We have added this note to the caption of Figure 5: "… averaged over the simulation period…".